# CORRECTING FLOWS WITH MARGINAL MATCHING

## ABSTRACT

Flow matching models, ODE-based generative models, generate samples by gradually morphing a simple source distribution into a target distribution. In practice, these models still fall short of perfectly replicating the target distribution, mainly due to imperfections of the learned mapping. Previous work mainly focus on alleviating discretization error, which rises from sampling a continuous trajectory with a finite number of steps. In this work we focus on prediction error, an error that is inherent in the model. Our main contribution is identifying a trajectory that complies with the imperfect flow model and leads exactly to the target distribution. Based on this finding, we propose Marginal Matching—a simple inference-time correction scheme to steer the generated samples in the direction of the data. This scheme proves to reduce a bound on the distance between the data and the learned distribution, motivating two different implementations for the correction function. We show that our proposed method improves sample quality on CIFAR-10 and ImageNet-64, with minimal overhead in computation time, or non at all when applying approximated correction.

## 1 INTRODUCTION

Flow based generative models (Lipman et al., 2023; Liu et al., 2023a; Tong et al., 2024) can generate complex data distributions, and have achieved remarkable results in image synthesis, audio generation, protein design and robotics (Yan et al., 2024; Liu et al., 2023b; Le et al., 2024; Irwin et al., 2024; Hu et al., 2023). These models learn a mapping between a source distribution (typically Gaussian noise) and a target distribution. In practice, however, this learned mapping is not accurate due to two main sources of error: discretization and prediction. The discretization error stems from the mapping, which is defined as a continuous time operator, with discrete steps, while the prediction error is attributed to general neural network training difficulties, such as limited architecture's expressivity, limited training time and numerical instability. Previous works focus on minimizing the discretization error, for example by learning straighter trajectories (Pooladian et al., 2023; Lee et al., 2023; Tong et al., 2024). In this work we focus on mitigating the prediction error of a pre-trained model, which, to the best of our knowledge, has not yet been tackled in the flow matching literature.

Flow matching models assume that there exists an ODE that maps Gaussian noise to the data distribution. Errors in learning this ODE result in an imperfect mapping, which leads to a generated distribution that is different from the data. In reversible models one can also consider starting from the data distribution and following the reverse mapping. The idea of approximating and implementing the reversal mapping has been explored before, in the context of diffusion models (Wallace et al., 2023; Mokady et al., 2023). Here, we exploit time reversibility for flow matching models, which is arguably less involved, as it only requires solving the ODE in reverse time (Liu et al., 2023a).

Our main insight is that starting the generation from samples on the reverse time trajectory, (with the data as initial distribution), will perfectly reach the target distribution under the model's mapping. We build on this insight and propose a simple method to improve the quality generation via correcting the sampled trajectory during inference. The correction is done by applying a learned correction step intermittently with the ODE solver steps, which works to reduce the error between the generated samples and the time-reversal ODE solution. We provide theoretical guarantees that such a correction function minimizes a bound on the Wasserstein distance (Kantorovich, 1942) between the target approximation and the true target distribution, as a function of the error reduction of the corrections at different time steps.

We examine two practical implementations for the correction function. The first implementation minimizes a theoretical bound, but has access to only subset of the data during training. The second has access to the full dataset, but makes no theoretical guarantee. We examine these correction models empirically and perform an extensive ablation study for our design choices. Our results show significant improvement with minimal additional steps, or with no extra compute time at all by running an approximated correction in parallel to the flow model.

In summary, our **main contributions** are as follows:

- A novel framework for improving pre-trained flow matching models using the concept of time reversal.
- A theoretical analysis of this general framework, and two practical implementations.
- We empirically show that our proposed correction models improve results on CIFAR-10 and ImageNet-64 datasets, achieving lower FID scores with fewer sampling steps.

## 2 BACKGROUND

### 2.1 FLOW MODELS

Given a source distribution $\pi_0$ and a target distribution $\pi_1$, both over $\mathbb{R}^d$, flow matching (Lipman et al., 2023; Liu et al., 2023a; Tong et al., 2024) models their mapping. This is done with a smooth time-dependent vector field $u : [0, 1] \times \mathbb{R}^d \to \mathbb{R}^d$ which defines an ODE $d\phi_t(x) = u_t(\phi_t(x))dt$, where $x \in \mathbb{R}^d$, $t \in [0, 1]$ and $\phi_t$ is the flow map with the initial condition $\phi_0(x) = x$. A time-varying density function $p : [0, 1] \times \mathbb{R}^d \to \mathbb{R}_{>0}$ that satisfies the continuity equation:

$$\frac{\partial p_t}{\partial t} + div(p_t(x)u_t(x)) = 0, \tag{1}$$

with the initial density $p_0$ over $\mathbb{R}^d$ is called the *marginal probability path* generated by the vector field $u$, and $u$ is said to be the *probability flow ODE* for the density function $p$.

The flow matching (FM) objective regresses $v_\theta : [0, 1] \times \mathbb{R}^d \to \mathbb{R}^d$, a time-dependent vector field parametrized as a neural network with weights $\theta$, to a target vector field $u_t$ via the following mean squared error (MSE) loss function:

$$\mathcal{L}_{FM}(\theta) = \mathbb{E}_{t \sim [0,1], x \sim p_t(x)} \|v_\theta(t, x) - u_t(x)\|^2. \tag{2}$$

Numerous mappings exist between probability distributions $\pi_0$ and $\pi_1$. The distributions $p_t(x)$ and $u_t(x)$ are not unique and typically impossible to compute directly. To overcome this challenge, Lipman et al. (2023) demonstrated that equivalent gradients to Eq. 2 with respect to $\theta$ can be derived using an alternative approach named conditional flow matching (CFM) loss:

$$\mathcal{L}_{CFM}(\theta) = \mathbb{E}_{t \sim [0,1], z \sim q(z), x \sim p_t(x|z)} \|v_\theta(t, x) - u_t(x|z)\|^2, \tag{3}$$

where $z$ is some conditional variable sampled from a distribution $q(z)$. The full derivation is available in Lipman et al. (2023); Tong et al. (2024). Many efficient parameterizations exist for $u_t(x|z)$ and $p_t(x|z)$, in this work we focus on the parametrization described in Tong et al. (2024):

$$z = (x_0, x_1) \quad ; \quad u_t(x|z) = x_1 - x_0 \quad ; \quad p_t(x|z) = \mathcal{N}(x|t \cdot x_1 + (1 - t) \cdot x_0, \sigma^2 I),$$

where $z \sim q(z)$ and $t \in [0, 1]$ is the interpolation coefficient. The conditional density $p_t(x|z)$ specifies one of the possible $p_t$ distributions and is easy to sample from and learn. The vector field $u_t$ and its corresponding marginal distribution $p_t$ can be obtained in terms of the conditional ones: $p_t(x) = \int p_t(x|z)q(z)dz$, $u_t(x) = \int \frac{p_t(x|z)u_t(x|z)}{p_t(x)}q(z)dz$. We examine two recently studied flow models by Tong et al. (2024), one with independent coupling (I-CFM) $q(z) = \pi_0(x_0) \times \pi_1(x_1)$ and another with optimal transport coupling (OT-CFM) $q(z) = OT(x_0, x_1)$. The optimal transport (OT) problem maps $x_0$ to $x_1$ such that a displacement cost, (typically the squared Euclidean distance), is minimized, for more details see Appendix. A.1. The OT coupling is calculated on batches during training and results in straighter trajectories, (Pooladian et al., 2023; Tong et al., 2024).

### 2.2 SCORE MODELS

In score-based generative models (Song et al., 2021b; Song & Ermon, 2019) the forward and reverse processes are modeled using standard diffusion and reverse-time stochastic differential equations

(SDEs). The forward process is described by $dx = f(x,t)dt + g(t)dw$, while the reverse-time SDE is given by $dx = \left(f(x,t) - g(t)^2\nabla\log\pi_1\right)dt + g(t)dw$. In these equations, $w$ represent standard Wiener processes, $f$ is the drift coefficient, $g$ is the diffusion coefficient, and $t$ ranges from 0 to $T_s$. The reverse SDE is used for generating samples, evolving from $t = T_s$ to $t = 0$, for example by using a numerical SDE solvers.

The score model estimates the score function $\nabla_x\log\pi_1(x)$, where $\pi_1(x)$ is the target distribution defined above. The training process typically involves denoising score matching (DSM):

$$L_{DSM}(\theta) = \mathbb{E}_{\pi_1(x)p_\sigma(\tilde{x}|x)}\|s_\theta(\tilde{x}) - \nabla_{\tilde{x}}\log p_\sigma(\tilde{x}|x)\|_2^2 \tag{4}$$

Here, $s_\theta$ denotes a neural network parameterized by $\theta$, and $p_\sigma(\tilde{x}|x) = \mathcal{N}(x, \sigma^2 I)$ represents the probability of a noisy sample from the target distribution. Vincent (2011) demonstrated that the optimal score network $s_{\theta*}(x)$, which minimizes Eq. 4, satisfies $s_{\theta*}(x) = \nabla_x\log p_\sigma(x)$ almost surely. This result shows that the network learns to estimate the score function of $p_\sigma(x)$, which is crucial for effectively denoising it. However, the approximation $s_{\theta*}(x) = \nabla_x\log p_\sigma(x) \approx \nabla_x\log\pi_1(x)$ holds true only when the noise level is sufficiently small, such that $p_\sigma(x) \approx \pi_1(x)$.

## 3 METHOD

### 3.1 PROBLEM FORMULATION AND MOTIVATION

Let $u^*$ denote an "ideal" flow function that maps the source distribution $\pi_0$ to the target distribution $\pi_1$. In this work we follow the formulation of Tong et al. (2024) and assume the conditional marginals' distributions specified by $u^*$ are $p_t(x|z) = \mathcal{N}(x|t\cdot x_1 + (1-t)\cdot x_0, \sigma)$[1]. Nonetheless, our derivations are not limited to this particular flow. We are given a pre-trained flow model, $v_\theta$, that was trained to approximate $u^*$. However, as is common in most practical applications $v_\theta \neq u^*$. Let $\hat{\pi}_1$ denote the distribution of samples generated by the model $v_\theta$. In practice, $\hat{\pi}_1$ will not exactly match the target distribution $\pi_1$ due to two primary sources of error:

- **Prediction error:** This error is inherent in the learned model, as $v_\theta \neq u^*$. The primary sources for this error are limited expressivity of the model's architecture, limited training time and numerical instability. In addition, this error is more pronounced in models in which sampling is iterative, as it accumulates with each forward iteration.
- **Discretization error:** This error arises from approximating a continuous-time trajectory with discrete steps. That is, even if $v_\theta = u^*$, it may be that $\hat{\pi}_1 \neq \pi_1$ due to the discretization in the numerical integration method.

Previous works tried to tackle the discretization error, for example by learning straighter trajectories (Pooladian et al., 2023; Tong et al., 2024; Lee et al., 2023). This reduces the discrepancy between the continuous trajectory and its discretized approximation. In contrast, the prediction error is more challenging to address and has remained relatively unexplored. In this work, we focus on this error; given $v_\theta$ and access to the data it was trained with, we seek to improve the performance of generating samples from $v_\theta$.

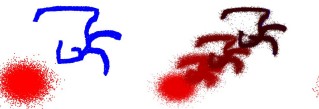
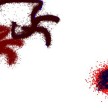

(a) $\pi_0$ and $\pi_1$     (b) Forward     (c) Backward

Figure 1: **Illustration of Marginals** (a) Gaussian source distribution $\pi_0$ (red), and target distribution $\pi_1$ (blue). (b) Discrete forward marginals $\{p_{t_n}^f\}_{n=0}^N$. (c) Discrete backward marginals $\{p_{t_n}^b\}_{n=0}^N$. Note that $p_{t_n}^b \neq p_{t_n}^f$ for every $n$, specifically $p_0^b$ (=$\hat{\pi}_0$, black) differs from $p_0^f$ (=$\pi_0$, red).

A straightforward idea to mitigate prediction error is to attempt to reduce the distance between $v_\theta$ and $u^*$. However, as this was already the training objective of $v_\theta$, it is not clear how to make a principled improvement in this direction. Instead, we propose a different idea, based on the following insight.

We observe that if the vector field represented by $v_\theta$ is invertible, a well defined "reverse flow" exists from the target distribution $\pi_1$ according to the reverse of the vector field $-v_\theta$. The key observation is that starting a forward flow using $v_\theta$ from any point on the reverse flow will converge to the target

---

[1]To keep it concise, we will henceforth use the notation $p_t(x)=p_t(x|z)$

distribution. That is, we have found a trajectory of densities that works perfectly with our imperfect model $v_\theta$. Our proposal is to learn how to correct the original trajectory density path to be more similar to this trajectory, with the hope that by doing so, applying the flow with $v_\theta$, will bring us closer to the target distribution. Thus, correcting deviations introduced by the prediction error.

We assume that the flow is invertible, implying cycle consistency between forward and backward vector field mappings (Zhu et al., 2017). A similar assumption is used in Liu et al. (2023a) in experiments on image translation. While we cannot verify that this assumption holds in practice, we empirically validate our method on popular datasets.

To explain our method, we next define the forward and backward marginals, which represent the evolving probability distributions over time generated by the vector field. These marginals will be a key component to understand how we adjust the sampling process.

**Definition 3.1.** *Forward and Backward Marginals: Let $u$ be a continuous globally Lipschitz time-dependent function $u : [0,1] \times \mathbb{R}^d \to \mathbb{R}^d$. Let $u^{-1}$ be the reverse vector field defined as: $u^{-1}(t,x) := -u(t,x)$. The discretization of the time-interval $[0,1]$ into $N$ steps is $t_n = n/N$.*

- *Forward Marginals are the probability path generated by $\frac{\partial p_t^f}{\partial t} + div(p_t^f(x)u_t(x)) = 0$ from time 0 to 1, with initial distribution $p_0^f = \pi_0$. The final distribution is $p_1^f = \hat{\pi}_1$. Their $N$ steps discretization is defined as $\{p_{t_n}^f\}_{n=0}^N$.*

- *Backward Marginals are the probability path generated by $\frac{\partial p_t^b}{\partial t} + div(p_t^b(x)u_t^{-1}(x)) = 0$ from time 1 to 0, with initial distribution $p_1^b = \pi_1$. The final distribution is $p_0^b = \hat{\pi}_0$. Their $N$ steps discretization is defined as $\{p_{t_n}^b\}_{n=0}^N$[2].*

We refer to $p_0^b$ as the approximate source distribution to distinguish it from $\pi_0$, the source distribution, and $p_1^f$ as the approximate target distribution to distinguish it from $\pi_1$, the target distribution. Note that $p_0^b$ is the optimal source distribution for $v_\theta$, in the sense that integrating over the vector field from $p_0^b$ would guarantee reaching the distribution $\pi_1$. Fig. 1 illustrates the forward and backward marginals of a flow model sampled with 10-steps Euler integration. It demonstrates the probability marginals trajectory and that $p_{t_n}^b \neq p_{t_n}^f$ for every $n$. Additionally, there is a substantial intersection between the forward and backward marginals, and their symmetric difference represents the outliers that do not reach the target distribution, for more details see Appendix. A.5.

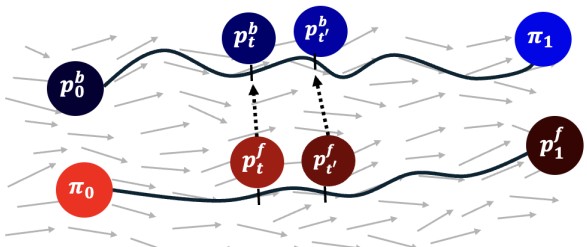

Figure 2: **Marginals in Vector Field** The black lines show two key trajectories in a vector field. The lower is the forward trajectory, starts at $\pi_0$ and ends at $p_1^f = \hat{\pi}_1$. The upper is the backward trajectory, begins at $p_0^b$ and ends at the data distribution $\pi_1$. Both are discretized at time-steps $t$ and $t'$. Our objective is to transition from forward to backward marginals trajectory.

Our next observation is that if the forward marginals trajectory reaches the target distribution, then it aligns with the backward trajectory for every time-step. This is formally stated in Lemma. 3.2

**Lemma 3.2.** *Let $u : [0,1] \times \mathbb{R}^d \to \mathbb{R}^d$ be a continuous time-dependent vector field, satisfying: $|u_t(x) - u_t(y)| \leq L|x-y|, \forall t \in [0,1], x,y \in \mathbb{R}^d$. Then, $p_1^f = p_1^b = \pi_1$ if and only if $p_t^f = p_t^b$ $\forall t \in [0,1]$.*

Based on this observation, we propose a novel approach to correct a flow model: rather than reducing the discrepancy between $v_\theta$ and $u^*$, we suggest reducing the discrepancy between the forward and backward marginals. We term this process Marginal-Matching (MM). Fig.2 illustrates this idea. The following lemma allows bounding the distance between the marginals at time $t$ based on the distance between the marginals at time $t_0$:

---

[2]To maintain clarity, we use the same time indexing rationale for all marginals. That is, a subscript 0 corresponds to the initial point of the forward marginals and the final point for the backward marginals.

**Lemma 3.3.** *Let $u$ be a vector field as defined in Lemma. 3.2. The 2-Wasserstein distance between $p_t^f$ and $p_t^b$ satisfies:*

$$W_2(p_t^f, p_t^b) \leq W_2(p_{t_0}^f, p_{t_0}^b) \exp\{L(t - t_0)\}.$$

This lemma indicates that reducing the Wasserstein distance at the initial $t_0$ can effectively control the Wasserstein distance at later times. We can apply this principle iteratively, by applying several corrections along the interval $[0, 1]$, and bounding the distance between the probability distributions on the continuous sub-intervals between each correction step. The next theorem bounds the accumulated error reduction due to applying such corrections. Let $h$ be a function from a family of functions that reduce the Wasserstein distance between the forward and backward marginals. We claim that applying any function from this family on the forward marginals during inference would improve generation.

**Theorem 3.4.** *(Informal) Let $u$ be a well-behaved vector field with Lipshitz constant L, $p_t^f$ and $p_t^b$ be two time-dependent probability density functions satisfying the continuity equation Eq. (1) with respect to $u$. Suppose that the initial Wasserstein distance is $d_0 = W_2(p_0^b, p_0^f)$. At each time step $t_n$, a correction function $h$ is applied to $p_{t_n}^f$ and the flow function continues from $h(t_n, p_{t_n}^f)$. Assuming $h$ reduces the $W_2$ distance to $p_{t_n}^b$ by $\epsilon_n$, the reduction on the bound of the final distance is: $W_2(p_1^b, p_1^f) \leq d_0 \exp\{L\} - \sum_{i=0}^{n-1} \epsilon_i \exp\{L(1 - t_i)\}.$*

A formal theorem statement and complete proofs and are in Appendix. A.2. In the appendix, we calculate the bound of Thm. 3.4 for two specific error reduction functions - additive and multiplicative. In the following examples, we demonstrate that in each case, the total error reduction is spread differently across the time steps, demonstrating an important idea – we should prioritize the corrections applied in particular steps based on the error reduction model. We will revisit this idea in our experiments.

**Example 3.5.** *Linear Reduction* This is the case of linear error reduction in *Thm. 3.4. Suppose $N = 2$, then the resulting bound takes the form: $d_0 \exp\{L\} - \epsilon_0 \exp\{L\} - \epsilon_1 \exp\{L \cdot 0.5\}$. Assuming $\epsilon_0 = \epsilon_1$, the reduction term at time step 0 exerts a more significant influence on the final bound due to the larger exponential factor. This underscores that, in such cases, the focus should be on prioritizing the correction of earlier steps in the sequence.*

**Example 3.6.** *Multiplicative Reduction* This is the case of multiplicative error reduction in *Thm. 3.4. Suppose $N = 2$, then the resulting bound takes the form: $d_0 \exp\{L\}(1 - \epsilon_0)(1 - \epsilon_1)$. In this formulation, the reduction terms at each time step $(1 - \epsilon_0)$ and $(1 - \epsilon_1)$ contribute equally to the final bound, regardless of their position in the sequence. This structure implies that the optimal strategy for minimizing the bound would be to prioritize improvements at the steps where they yield the greatest benefit—specifically, where the magnitude of $\epsilon_i$ is largest.*

### 3.2 PRACTICAL ALGORITHM

Our goal is to correct the sampled trajectory during inference to reduce the discrepancy between the forward and backward marginals at every step. This section proposes such an algorithm given a pre-trained flow model $v_\theta$. The flow model is designed to transform an easily sampled source distribution $\pi_0$, (such as $\mathcal{N}(0, I)$), into a target distribution $\pi_1$. Rather than considering only the approximate source distribution ($p_0^b$), motivated by Thm. 3.4, our approach considers all backward marginals.

---

**Algorithm 1** Corrected Inference

**Require:** flow model $v_\theta$, correction model $h_\psi$, number of iterations $N$, step size of corrector steps $\{\alpha_i\}_{i=0}^N$, scale added noise $\{\beta_i\}_{i=0}^N$
1: $x_0 \sim \mathcal{N}(0, I)$
2: **for** $n = 0, \ldots, N - 1$ **do**:
3:    $\epsilon \sim \mathcal{N}(0, I)$
4:    $\tilde{x}_{t_n} = x_{t_n} + \beta_n \cdot \epsilon$
5:    $x_{t_n} = x_{t_n} + \alpha_n \cdot h_\psi(t_n, \tilde{x}_{t_n})$  ▷ Correction
6:    $x_{t_{n+1}} = x_{t_n} + \frac{1}{N} v_\theta(t_n, x_{t_n})$   ▷ Flow
7: **end for**
8: $x_{t_N} = x_{t_N} + \alpha_N \cdot h_\psi(t_N, x_{t_N})$  ▷ Correction
9: **return** $x_{t_N}$

---

Recall from Thm.3.4 that the transport map $h$ can be utilized to reduce the distance between the forward $p_t^f$ and backward marginals $p_t^b$. In the following, slightly abusing notation, we assume that $h$ is the push-forward of $x \in \mathbb{R}^d$ with a mapping $h(x)$. In practice, the specific error reduction is unknown and we can only require that the bound on the Wasserstein distance will decrease after $h$ is applied. Let $h_\psi$ be a neural network with weights $\psi$ that approximates the correction function $h$. Algorithm 1 presents our proposed approach, which

enhances the inference process. It introduces correction steps using $h_\psi$ before each step of the flow model $v_\theta$ and at the end. For clarity, the algorithm assumes Euler integration for sampling $v_\theta$ and single correction steps, though multiple corrections per time-step are possible. Adding Gaussian perturbation before each correction step, except the last one, improves performance.

**Parallel Sampling:** The corrected inference process, as described in Algorithm 1, introduces additional sampling time for each correction step. To reduce the overhead, we propose a correction algorithm that is parallelizable in principle, by running the correction and the flow model in parallel. The correction and the flow model both operate on the same input at the same time, then the correction $h_\psi$ and the flow model's $v_\theta$ direction are summed, approximating the full correction process.

While this is an approximation, the effectiveness of this approximation relies on the assumption that $h_\psi$ and $v_\theta$ do not change too rapidly. The full algorithm is in Appendix. A.3.1.

In the following we consider two models for implementing the correction model $h_\psi$.

### 3.2.1 SCORE MODEL

Our first proposed approach is a time-dependent score model $h_\psi(t, x_t) = s_\psi(t, x_t)$, presented in Algorithm. 2, to approximate the backward marginals $\nabla_{x_t} \log p_t^b(x_t)$, $x_t \in p_t^f$. We assume the forward marginals are their

---

**Algorithm 2** Score Model Training

**Require:** flow model $v_\theta$, score model $h_\psi$, $\sigma$, backward marginals $\{p_{t_n}(x)^b\}_{n=0}^N$

1: **repeat**
2:     $\epsilon \sim \mathcal{N}(0, I)$
3:     $n \sim U(\{0, 1, .., N\})$
4:     $x_{t_n} \sim p_{t_n}^b(x)$
5:     Take gradient descent step on
6:     $\nabla_\psi \|h_\psi(t_n, x_{t_n} + \sigma \cdot \epsilon) + \epsilon\|^2$
7: **until** converged

---

"noisy" versions, i.e. $p_t^f = p_{t,\sigma}^b$ where $p_{t,\sigma}^b(\tilde{y}_t | y_t) = \mathcal{N}(y_t, \sigma^2 I)$ and $y_t \sim p_t^b$. Motivated by the bound of Kwon et al. (2022) (Thm. 2) for score models generation, the following lemma bounds the final Wasserstein distance after applying the score correction on a single time-step marginal:

**Lemma 3.7.** *(Informal) Applying $s_\psi$ from $p_{t_n}^f$ to approximate $p_{t_n}^b$ as described in Sec. 2.2 bounds the final distance $W_2(p_1^b, p_1^f)$ as follows:*

$$W_2(p_1^b, p_1^f) \leq C(s_\psi) \exp\{L(1 - t_n)\},$$

where $C(s_\psi)$ depends on the score model's loss $L_{DSM}^\psi$ as defined in Sec. 2.2. For small enough $L_{DSM}^\psi$, this bound improves upon using no correction. For more details and a bound on the general case—applying score correction on multiple marginals, see Appendix. A.2.

### 3.2.2 ROBUST CLASSIFIER

In the score model above, we did not use data from the forward marginals during training (we treated them as "noisy" versions of the backward marginals). We next propose a method that explicitly transitions between samples from the forward and backward marginals. This approach utilizes a time-dependent classifier $c_\psi(t, x_t)$ that distinguishes between forward and backward marginals ($p_t^f$ and $p_t^b$), evaluating each sample independently. By learning to categorize samples as belonging to either the forward or backward marginal at each time step, the classifier's gradients $h_\psi(t, x_t) = \nabla_{x_t} \log c_\psi(t, x_t)$ provide corrective guidance, effectively steering samples toward the backward marginals. The gradients will be zero upon reaching the correct class.

To enhance the classifier's gradient performance, we implement two key improvements: Adversarial Training (**AT**) and Gradient Alignment (**GA**). These keep the classifier's gradients stable and meaningful, for more details and the complete algorithm for training see Appendix. A.3.3.

**Usage for Class Conditioning** Classifier guidance is used in diffusion models and more recently in flow models (Sun et al., 2024) in order to generate conditional images. So far, MM is used to generate samples from the data, however, we can extend this method to also generate conditional data. For this purpose, our classifier can be conditioned $c_\psi(t, x_t, l)$ where $l$ is a class label, and trained to categorize different classes in the backward trajectory. For an illustration see Appendix. A.6.2.

## 4 RELATED WORK

**Training and inference mismatch** – Previous work recognized training and inference mismatch in different settings. In score models (Song et al. (2021b); Song & Ermon (2019)), the source

distribution during inference is $\mathcal{N}(0, I)$, which is intended to approximate the "noisiest" training distribution ($p_T$). However, as these distributions are not identical, starting from $\mathcal{N}(0, I)$ will yield sub-optimal results. Two different solutions were proposed to minimize this gap as a pre-sampling procedure: Franzese et al. (2023) suggested an auxiliary model that learns the mapping from $\mathcal{N}(0, I)$ to $p_T$. Alternatively, Pedrotti et al. (2024) proposed to use Langevin dynamics (Welling & Teh (2011)), leveraging the score function $\nabla_x \log p_T(x)$.

Even if the source distribution during training and inference is the same, practically, the actual distribution that reaches the data is different. Coeurdoux et al. (2023) leveraged the invertibility property of normalizing flow (NF) models (Kingma & Dhariwal (2018); Dinh et al. (2014)) to reach higher probability samples in $\hat{\pi}_1$, which are more likely to belong to $\pi_1$. They achieved this by employing MCMC algorithm, where the score is computed using the Jacobian of the inverse mapping. This discrepancy between the learned and actual source distribution is common in generative models like VAEs (Kingma et al., 2021), which assume a Gaussian prior during inference. (Dai & Wipf, 2019) addressed this by learning an additional VAE model to predict the actual prior distribution, thereby removing the Gaussian assumption, which lead to improved results.

**Different sampling trajectories** – Different sampling trajectories were explored in previous work mainly through the research of difference source distributions. In Denoising Diffusion Probabilistic Models (Ho et al. (2020); Sohl-Dickstein et al. (2015)), various source distributions proved to decrease inference time while maintaining quality. gil Lee et al. (2022) proposed using a Gaussian prior distribution with parameters derived from data statistics. Lyu et al. (2022); Popov et al. (2021) trained a model for prior distribution using an encoder or VAE (Kingma & Welling (2013)) and incorporated it into the standard diffusion process by adding noise. Other studies have shown that the prior distribution need not be Gaussian Bansal et al. (2024); Heitz et al. (2023). Flow based models Lipman et al. (2023); Tong et al. (2024); Liu et al. (2023a), a new family of generative models, generalized the mapping to include general source distributions.

The work most closely aligned with ours is that of Xu et al. (2024) as they considered the learned errors throughout the whole sampling trajectory. They use a sequence of NF blocks to learn a sequential transformation from Gaussian noise to data, where each block represents a different time-step. This requires training the blocks sequentially, as the learned error in each block is accounted for in the next block, slowing the training process. The sampling process initiates from Gaussian noise, where the model's error is not accounted for. In contrast, our work takes into account the model's prediction error at every step, as we correct the inference time trajectory. Moreover, we improve the recently proposed flow matching instead of NF. For extended related work see Appendix. A.4.

## 5 EXPERIMENTS

We empirically evaluate MM's image generation performance, with both the score and classifier correction models. The score model is trained with a constant noise $\sigma$. We conduct experiments on two datasets: unconditional CIFAR-10 (Krizhevsky et al. (2009)) and ImageNet-64 (Chrabaszcz et al. (2017); Deng et al. (2009)). To assess the quality of generated images, we employ Fréchet Inception Distance (FID) (Heusel et al. (2017)) score as our evaluation metric.

A key factor in diffusion and flow models is inference compute time, measured by number of function evaluations (NFE). In all tables, NFE represents the total number of sampling steps, and C-NFE denotes the number of sampling steps taken with our correction model (out of the NFE). We note that a classifier correction step require two NFEs, due to input derivative calculations, while score correction requires only one NFE.

For the flow model we use the UNet architecture with the same hyper-parameters as Tong et al. (2024). For the score and classifier models, we use UNet with half the number of parameters used for the flow model. Additionally, these correction models undergo significantly fewer training iterations compared to the flow model. Unless otherwise specified, the correction models were trained on trajectories consisting of $N = 11$ marginals with OT-CFM (Tong et al., 2024) as the flow model, and the flow model was sampled with 10-step RK4, which is 40 NFEs. For more implementation details, please refer to the Appendix. A.10. The code will be available upon publication.

| Model | Sampler | NFE↓ | FID↓ | Ours | NFE↓ | C-NFE↓ | OT-CFM FID↓ | I-CFM FID↓ |
|---|---|---|---|---|---|---|---|---|
| DDPM | Adaptive | 274 | 7.48 | Score | 41 | 1 $(n=10)$ | 3.45 | 3.47 |
| Score Matching | Adaptive | 242 | 19.94 | | 43 | 3 $(n=0,5,10)$ | 3.38 | 3.39 |
| OT-FM | Adaptive | 142 | 6.35 | | 51 | 11 $(n=0\dots10)$ | **3.37** | **3.38** |
| OT-CFM | RK4 | 40 | 4.34 | Classifier | 42 | 1 $(n=8)$ | 3.57 | 3.77 |
| | Adaptive | 133.94 | 3.58 | | 46 | 3 $(n=8,9,10)$ | 3.48 | 3.67 |
| I-CFM | RK4 | 40 | 4.29 | | 50 | 5 $(n=6,7,8,9,10)$ | **3.47** | **3.62** |
| | Adaptive | 146.42 | 3.66 | | 62 | 11 $(n=0\dots10)$ | 3.48 | 3.63 |

Table 1: CIFAR-10 performance. (Left) Leading baseline models. (Right) Our correction models (score and classifier) with OT-CFM and I-CFM base flow models. The base flow models are sampled with 10-step RK4 (40 NFE). Correction steps improve performance with minimal additional NFEs, and generally, increasing the number of applied corrections (C-NFE) enhances overall results. Additional comparisons are available in Appendix. A.6.6.

## 5.1 CIFAR-10 SAMPLE QUALITY

To showcase the improvement of MM we implement our score and classifier correction models with OT-CFM and I-CFM base flow models on CIFAR-10. Table 1 shows comparison of our results to (Ho et al., 2020; Song et al., 2021b; Lipman et al., 2023; Tong et al., 2024). Previous works employ an adaptive sampler (DOPRI5, dor (1980)), while we use RK4 as the ODE-sampler to ease the integration of the correction steps. We compare with results reported in Lipman et al. (2023), implemented using the same UNet (Dhariwal & Nichol, 2021), and those in Tong et al. (2024). As Tong et al. (2024) did not report RK4 sampling results, we conduct these experiments and report baseline results using the models' checkpoints as our foundational flow models. Following the results of Sec. 3.1, we investigate applying the correction on various time steps (C-NFE > 1)[3]. We examine several options: for the score model the correction steps were taken on $n = \{[10], [0,5,10], [0\dots10]\}$ and for the classifier on $n = \{[8], [8,9,10], [6,7,8,9,10], [0\dots10]\}$.

We significantly outperform the baselines in the small NFE regime. The score model slightly outperforms the classifier in FID score, although it was not exposed to the forward marginals during training. Moreover, the score model demonstrates consistent performance regardless of whether it is applied to OT-CFM or I-CFM, indicating a robust ability to denoise from forward to backward marginals across both frameworks. Whereas, the robust classifier exhibits sensitivity to the choice of the model, showing superior performance when coupled with OT-CFM. We did not observe any benefit from applying both the score and classifier correction models together.

Fig. 4 (Left) presents a qualitative comparison of the correction performed by each algorithm, (additional images are available in Appendix A.8). The correction models improve clarity and sometimes make significant changes to the image. The score and classifier corrections show varying levels of effectiveness depending on the specific image. In some cases, one method outperforms the other, while in other instances, they surprisingly produce similar results.

**Parallel Sampling** Fig. 3 presents parallel sampling results as described in Sec. 3.2 on CIFAR-10. Both the classifier and score correction models show improvement in FID, though less than the exact correction version.

**Interpolating Marginals during Training** The correction model $h_\psi(t, x_t)$ is time-dependent, as it is trained on different time-step marginals. The marginals are generated and stored in advance to avoid running the ODE-solver multiple times while training $h_\psi(t, x_t)$. In order to save this extra storage, we propose to approximate the backward trajectory. Assuming the flow model's trajectories are sufficiently straight, we can interpolate between $p_0^b$, the approximate source distribution, and $p_1^b$, the target distribution by: $\hat{p}_{t_n}^b = t_n \cdot p_1^b + (1 - t_n) \cdot p_0^b$. This allows for efficient training storage as only the data and the approximate source distribution are stored rather than the entire backward trajectory. This means that the MM's dataset size is twice rather than $N$ times the size of the original data. Fig. 3 presents the results of the interpolated backward trajectory. The score model's performance remains steady, while the classifier exhibits degradation in its performance.

---

[3]Multiple corrections per time-step did not improve the score model but helped the classifier. However, for simplicity we apply one correction step per marginal.

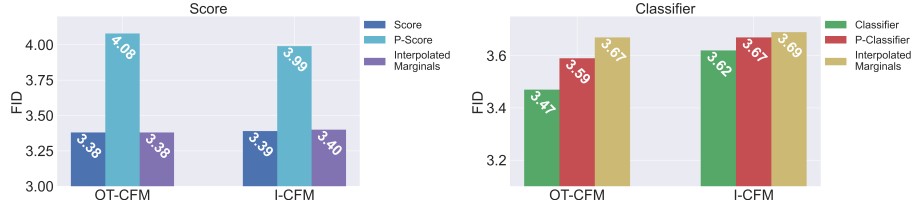

Figure 3: CIFAR-10 sample quality with different approximations – parallel ("P-") for faster inference and interpolated backward marginals for efficient storage during training; see text for details. The base flow is sampled with 10-step RK4 (40 NFE). For additional results, see Appendix. A.6.3.

| Model | NFE ↓ | Sample | FID ↓ |
|---|---|---|---|
| DDPM | 264 | Adaptive | 17.36 |
| Score Matching | 441 | Adaptive | 19.74 |
| OT-FM | 138 | Adaptive | 14.45 |
| OT-CFM | 20 | RK4 | 17.99 |
| | 24 | RK4 | 16.60 |
| | 28 | RK4 | 15.71 |
| | 32 | RK4 | 15.11 |

| Ours | NFE ↓ | C-NFE ↓ | FID |
|---|---|---|---|
| | 21 | 1 ($n=5$) | 15.91 |
| Score | 22 | 2 ($n=0,5$) | **15.53** |
| | 26 | 6 ($n=0\ldots5$) | 15.57 |
| Score | 21 | 1 ($n=5$) | 15.89 |
| (Backward Marginals | 22 | 2 ($n=0,5$) | **15.62** |
| Interpolation) | 26 | 6 ($n=0\ldots5$) | 15.69 |

Table 2: ImageNet-64 results for top generative models (Left) vs. score correction with OT-CFM base flow (Right), which is sampled with 5-step RK4 (20 NFE).

## 5.2 IMAGENET-64 SAMPLE QUALITY

We perform experiments on ImageNet-64 to examine how MM performs in a higher-dimensional settings. Given a pre-trained OT-CFM flow model trained on ImageNet-64, we train a score correction model to improve its sample quality, as the score model outperform the classifier on CIFAR, we chose to perform these resource-intensive experiments with it. The score model was trained on trajectories consisting of $N=6$ marginals. Table. 2 presents the improvement in FID of a correction model trained on the backward marginals trajectory and its approximation (for more details see Sec. 5.1) compared to (Ho et al., 2020; Song et al., 2021b; Lipman et al., 2023; Tong et al., 2024). Similarly to the CIFAR-10 results, we observe that MM improves FID in the low NFE regime (22 and below). The correction is qualitatively demonstrated in Fig. 4 (Right), (for more illustrations see Appendix. A.7). The correction model removes the residual noise in the model's approximation and changes the images structure to better align with the ImageNet dataset photos.

## 5.3 ABLATION STUDY

For more ablation studies, including varying $\sigma$ of the score correction model, see Appendix. A.6.1.

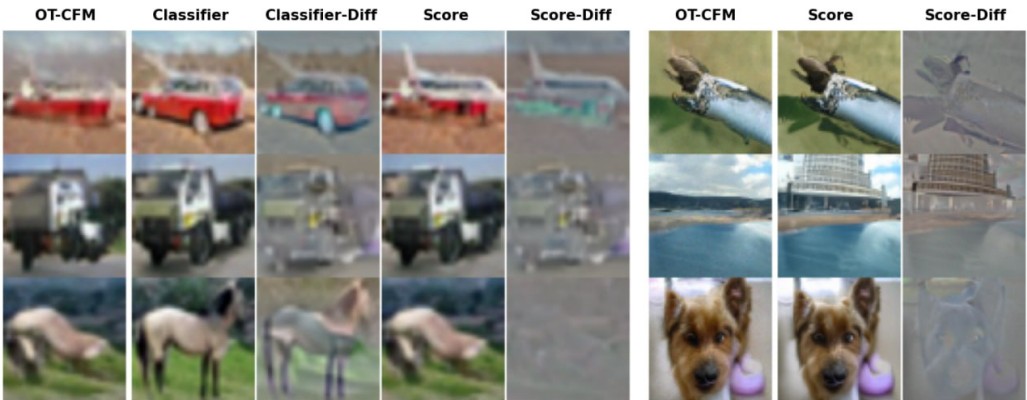

Figure 4: Correction models with OT-CFM as base flow model results on CIFAR-10 (Left) and Imagenet-64 (Right). The "-Diff" shows the difference between corrected and uncorrected images. Corrected images show improved sharpness and better alignment with the dataset.

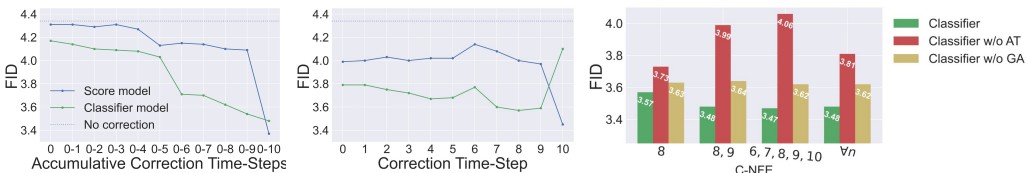

Figure 5: CIFAR-10 sample quality. The base flow model is OT-CFM sampled with 10-step RK4 (40 NFE). **Left:** FID vs accumulative correction time-steps. **Middle:** FID vs single correction time-step. **Right:** Ablation study on loss components of the classifier sampled on different marginals.

### 5.3.1 MARGINALS ABLATION

In this ablation we study the effect of applying correction on different time-steps. Fig. 5 (Left) demonstrates the cumulative effect of correcting all time-step marginals (from 0 to 10). The score model demonstrates significant improvements primarily during the middle and final steps, with minimal contributions from early stages. In contrast, the classifier exhibits a more uniform pattern of enhancement. Whereas, Fig. 5 (Middle) presents the improvement in FID when applying a single correction step on different time-step marginals. The classifier model's correction shows greater improvement than the score model at every time-step, with the exception of the final one. As predicted by the theoretical analysis in Sec. 3.1, we observe that different correction algorithms yield varying levels of improvement when applied at different time steps.

Notably, early-stage corrections mainly affect low-frequency components, shaping overall image structure. In contrast, later-stage corrections primarily impact high-frequency details, addressing noise-like elements (Kim et al., 2024). The striking impact of executing the last time step with the score model suggests residual noise in the model's predictions (see Appendix. A.9 for illustrations).

### 5.3.2 CLASSIFIER LOSS ABLATION

To assess the significance of each improvement made to the classifier, namely Adversarial Training (AT) and Gradient Alignment (GA), we conduct an ablation study by removing these components. The results are presented in Fig. 5 (Right). In the absence of AT, the gradients become unstable, which is evident from the inconsistent improvement observed when adding steps. On the other hand, without GA, the gradients are small, and adding steps does not impact the overall improvement.

## 6 CONCLUSION AND LIMITATIONS

Our main insight is that given an imperfect flow model, a trajectory that reaches exactly the data distribution can be computed by reversing its vector field. Based on this insight, we propose a simple algorithm, Marginal Matching, which steers the inference-time trajectory to better align with the trajectory that accurately reaches the data. We demonstrate superior performance on two datasets CIFAR-10 and ImageNet-64, and perform an extensive ablation study.

Ideally, all generative models would benefit from correction of prediction errors, as such errors are not specific to flow models. A major limitation of our work is the assumption on the reversibility of the vector field, constraining our method from operating on non-reversible models. This limitation may be relaxed for approximately reversible models such as diffusion models (Wallace et al., 2023). In general, extending our approach to non-reversible models is an interesting future direction.

Another aspect is the practical implementation of our theory. In Thm. 3.4 we assumed knowledge of error reductions. In practice, we do not have access to the model's error reduction (see Sec.3.1). However, we can bound that reduction in the case of score correction (Lemma. 3.7) and propose a bound on the final Wasserstein distance. Future work could explore theoretical bounds also for our classifier-based correction model.

Finally, our method requires training an additional model after a flow model has been pre-trained, adding extra compute and parameters. It is worth noting that in all our experiments, this auxiliary model is smaller than the original flow model, and training it takes significantly less time. An alternative approach could involve training the flow and correction models together, either as a single model with both flow and correction outputs or as two models trained simultaneously.

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

# A APPENDIX

## A.1 OPTIMAL TRANSPORT

Optimal Transport (OT) is a mathematical framework that addresses the problem of efficiently moving probability mass from one distribution to another while minimizing a certain cost. The central idea is to find the optimal way to transform one probability distribution into another, considering the "cost" of moving each unit of mass.

In the context of flow matching, before learning the flow, an optimal transport map is computed between batches of samples from $\pi_0$ and $\pi_1$. This OT step uses the 2-Wasserstein distance as its metric, with a cost function defined as the Euclidean distance $\|x - y\|$ between points. The flow is then learned based on these optimally transported samples, resulting in straighter trajectories (Pooladian et al., 2023; Tong et al., 2024).

## A.2 THEORETICAL PROOFS

*Proof of Lemma 3.2.* If $p_t^f = p_t^b$ for every $t$ then $p_1^f = p_1^b = \pi_1$ is trivial.

Assume $p_1^f = p_1^b = \pi_1$. We need to show that $p_t^f = p_t^b$ for every $t \in [0, 1]$.

The flow $\phi_t$ is a diffeomorphism for each $t \in [0, 1]$ due to the properties of the vector field $u$ (continuous and globally Lipschitz).

For any $t \in [0, 1]$, we can write:

$$p_t^f = [\phi_t]_\# p_0^f = [\phi_t \circ \phi_1^{-1}]_\# p_1^f,$$
$$p_t^b = [\phi_t]_\# p_0^b = [\phi_t \circ \phi_1^{-1}]_\# p_1^b,$$

where $\phi_1^{-1}$ is the inverse map of $\phi_1$ flowing from $t = 1$ to $t = 0$. Since the following equality holds: $p_1^f = p_1^b = \pi_1$, we can replace $p_1^f$ and $p_1^b$ with $\pi_1$ in the above equations: $p_t^f = [\phi_t \circ \phi_1^{-1}]_\# \pi_1$ $p_t^b = [\phi_t \circ \phi_1^{-1}]_\# \pi_1$.

$\square$

**Lemma A.1.** *[Lemma. 3.3 in the main paper] Let $u$ be defined as in Lemma. 3.2. The 2-Wasserstein distance between $q_t^1$ and $q_t^2$ satisfies:*

$$W_2(q_t^1, q_t^2) \leq W_2(q_0^1, q_0^2) \exp\{Lt\}.$$

**Corollary A.1.1.** *Assume $t' > t$: $W_2(q_{t'}^1, q_{t'}^2) \leq W_2(q_t^1, q_t^2) \exp\{L(t' - t)\}$.*

*Proof.* Let $\phi_t$ be the flow map of the vector field $u$ that induces the push-forward $q_t^1 := [\phi_t]_\# q_0^1$, $q_t^2 := [\phi_t]_\# q_0^2$ for any two probability density distributions $q_0^1$ and $q_0^2$ over $\mathbb{R}^d$, where $t \in [0, 1]$.

Let $\tilde{o}_0$[4] be the optimal coupling between $q_0^1$ and $q_0^2$. Denote the push-forward of $\tilde{o}_0$ as $\tilde{o}_t^\# := [\phi_t]_\# \tilde{o}_0$. Then:

---

[4]Literature commonly uses the symbol $\pi$ to denote an optimal transport coupling. This paper, however, already uses $\pi$ to represent source and target distributions. To avoid confusion, $\tilde{o}$ is chosen for the optimal transport coupling.

$$\frac{d}{dt} \int_{\mathbb{R}^d \times \mathbb{R}^d} \|x - y\|^2 d\tilde{o}_t^\#(x, y) = \frac{d}{dt} \int_{\mathbb{R}^d \times \mathbb{R}^d} \|\phi_t(x) - \phi_t(y)\|^2 d\tilde{o}_0(x, y) \tag{5}$$

$$= \int_{\mathbb{R}^d \times \mathbb{R}^d} \frac{d}{dt} \|\phi_t(x) - \phi_t(y)\|^2 d\tilde{o}_0(x, y) \tag{6}$$

$$= \int_{\mathbb{R}^d \times \mathbb{R}^d} 2(\phi_t(x) - \phi_t(y)) \cdot (\frac{d}{dt}\phi_t(x) - \frac{d}{dt}\phi_t(y)) d\tilde{o}_0(x, y) \tag{7}$$

$$= 2 \int_{\mathbb{R}^d \times \mathbb{R}^d} (\phi_t(x) - \phi_t(y)) \cdot (u_t(\phi_t(x)) - u_t(\phi_t(y))) d\tilde{o}_0(x, y)$$
$$\tag{8}$$

$$\leq 2L \cdot \int_{\mathbb{R}^d \times \mathbb{R}^d} \|\phi_t(x) - \phi_t(y)\|^2 d\tilde{o}_0(x, y) \tag{9}$$

$$= 2L \cdot \int_{\mathbb{R}^d \times \mathbb{R}^d} \|x - y\|^2 d\tilde{o}_t^\#(x, y), \tag{10}$$

where in (5) we changed variables, in (6) we used Leibniz's rule and in (8) the Lipshitz constraint. Using Grönwall's inequality, we have:

$$\int_{\mathbb{R}^d \times \mathbb{R}^d} \|x - y\|^2 d\tilde{o}_t^\#(x, y) \leq \int_{\mathbb{R}^d \times \mathbb{R}^d} \|x - y\|^2 d\tilde{o}_0(x, y) \exp\{2Lt\} \tag{11}$$

$$= W_2^2(q_0^1, q_0^2) \exp\{2Lt\}. \tag{12}$$

By the definition of Wasserstein distance:

$$W_2^2(q_t^1, q_t^2) \leq \int_{\mathbb{R}^d \times \mathbb{R}^d} \|x - y\|^2 d\tilde{o}_t^\#(x, y). \tag{13}$$

Substituting the bound:

$$W_2^2(q_t^1, q_t^2) \leq W_2^2(q_0^1, q_0^2) \exp\{2Lt\}. \tag{14}$$

Taking the square root of both sides yields the result:

$$W_2(q_t^1, q_t^2) \leq W_2(q_0^1, q_0^2) \exp\{Lt\}.$$

$$\square$$

**Theorem A.2.** *[Thm.3.4 in the main paper] Let $\mathcal{P}_2(\mathbb{R}^d)$ be the space of probability densities on $\mathbb{R}^d$ with finite second moments. Let $u$ be defined as in Lemma. 3.2. Let $\mu_t$ and $\nu_t$ be two time-dependent probability density functions in $\mathbb{R}^d$ satisfying the continuity equation Eq. (1) with initial densities $\mu_0$, $\nu_0$ and terminal densities $\mu_1$, $\nu_1$. Suppose that the initial 2-Wasserstein distance between them is $d_0 = W_2(\mu_0, \nu_0)$. The time interval $[0, 1]$ is discretized into $N$ equal steps: $t_n = n/N$, where $n = 0, 1, ..., N$. At each time step $t_n$, a time-dependent transport map $h : [0, 1] \times \mathcal{P}_2(\mathbb{R}^d) \to \mathcal{P}_2(\mathbb{R}^d)$ is applied to $\nu_{t_n}$. After applying $h$, the flow function at time $t_{n+1}$ continues from the probability density of $h(t_n, \nu_{t_n})$. We consider two variations for $h$:*

*A If $h$ reduces the 2-Wasserstein distance by a constant amount $\epsilon_n > 0$ at each step: $W_2(\mu_{t_n}, h(t_n, \nu_{t_n})) = W_2(\mu_{t_n}, \nu_{t_n}) - \epsilon_n$. Then, the following bound hold for $n = 0, 1, ..., N$: $W_2(\mu_{t_n}, \nu_{t_n}) \leq d_0 \exp\{L \cdot t_n\} - \sum_{i=0}^{n-1} \epsilon_i \exp\{L(t_n - t_i)\}$,*

*B If $h$ reduces the 2-Wasserstein distance to a proportion of the current distance: $W_2(\mu_{t_n}, h(t_n, \nu_{t_n})) = (1 - \epsilon_n) \cdot W_2(\mu_{t_n}, \nu_{t_n})$ where $0 < \epsilon_n < 1$ for all $n$. Then, the following bound hold for $n = 0, 1, ..., N$: $W_2(\mu_{t_n}, \nu_{t_n}) \leq d_0 \cdot \prod_{i=0}^{n-1} (1 - \epsilon_i) \cdot \exp\{L \cdot t_n\}$.*

*Proof of Theorem A.2.* We will prove parts A and B of the theorem separately.

**Part A:**

Let $d_n = W_2(\mu_{t_n}, \nu_{t_n})$. We will prove by induction that:

$$d_n \leq d_0 \exp\{L \cdot t_n\} - \sum_{i=0}^{n-1} \epsilon_i \exp\{L(t_n - t_i)\}.$$

Base case ($n = 0$): Trivially true as $d_0 = W_2(\mu_0, \nu_0)$.

Inductive step: Assume the inequality holds for $n$. We'll prove it for $n + 1$.

After applying $h$ at $t_n$, we have:

$$W_2(\mu_{t_n}, h(t_n, \nu_{t_n})) = d_n - \epsilon_n.$$

By Lemma. A.1 over the interval $[t_n, t_{n+1}]$, we have:

$$d_{n+1} \leq (d_n - \epsilon_n) \exp\{L(t_{n+1} - t_n)\}.$$

Substituting the inductive hypothesis:

$$d_{n+1} \leq \left( d_0 \exp\{L \cdot t_n\} - \sum_{i=0}^{n-1} \epsilon_i \exp\{L(t_n - t_i)\} - \epsilon_n \right) \exp\{L(t_{n+1} - t_n)\}$$

$$= d_0 \exp\{L \cdot t_{n+1}\} - \sum_{i=0}^{n-1} \epsilon_i \exp\{L(t_{n+1} - t_i)\} - \epsilon_n \exp\{L(t_{n+1} - t_n)\}$$

$$= d_0 \exp\{L \cdot t_{n+1}\} - \sum_{i=0}^{n} \epsilon_i \exp\{L(t_{n+1} - t_i)\}.$$

This completes the induction. The final bound at $t = 1$ follows by setting $n = N$.

**Part B:**

Again, let $d_n = W_2(\mu_{t_n}, \nu_{t_n})$. We will prove by induction that:

$$d_n \leq d_0 \cdot \prod_{i=0}^{n-1} (1 - \epsilon_i) \cdot \exp\{L \cdot n/N\}.$$

Base case ($n = 0$): Trivially true.

Inductive step: Assume the inequality holds for $n$. We'll prove it for $n + 1$.

After applying $h$ at $t_n$, we have:

$$W_2(\mu_{t_n}, h(t_n, \nu_{t_n})) = (1 - \epsilon_n) \cdot d_n.$$

By Lemma. A.1, over the interval $[t_n, t_{n+1}]$ we have:

$$d_{n+1} \leq (1 - \epsilon_n) \cdot d_n \cdot \exp\{L(t_{n+1} - t_n)\} = (1 - \epsilon_n) \cdot d_n \cdot \exp\{L/N\}.$$

Substituting the inductive hypothesis:

$$d_{n+1} \leq (1 - \epsilon_n) \cdot \exp\{L/N\} \cdot d_0 \cdot \prod_{i=0}^{n-1} (1 - \epsilon_i) \cdot \exp\{L \cdot n/N\}$$

$$= d_0 \cdot \prod_{i=0}^{n} (1 - \epsilon_i) \cdot \exp\{L \cdot (n+1)/N\}.$$

This completes the induction. The final bound at $t = 1$ follows by setting $n = N$.

$\square$

**Score Wasserstein Bound:**

The score-based models losses are:

$$L_{SM}(\theta; \lambda) := \frac{1}{2} \int_0^{T_s} \lambda(\tau) \mathbb{E}_{p_\tau} [\|\nabla \log p_\tau(x) - s_\theta(x, \tau)\|^2] d\tau,$$

$$L_{DSM}(\theta, \lambda) := \frac{1}{2} \int_0^{T_s} \lambda(\tau) \mathbb{E}_{p_0(x(0))p_{0\tau}(x|x(0))} [\|s_\theta(x, \tau) - \nabla_x \log p_{0\tau}(x|x(0))\|^2] d\tau,$$

where $\lambda : [0, T_s] \to (0, \infty)$ is a positive weighting function. The score model is typically trained using the widely adopted loss function known as $L_{DSM}$.

**Definition A.3.** *Noisy Backward Marginal: Let $p_{n,\sigma}^b$ be the noisy $n^{th}$ backward marginal. Then,*
$$p_{n,\sigma}^b(\tilde{y}_n|y_n) = \mathcal{N}(y_n, \sigma^2 I), \quad y \sim p_n^b.$$

**Theorem A.4** (Kwon et al. (2022)). *If $p_{0\tau}$ satisfies:*
$$Var[\mathbb{E}[(\nabla_x \log p_{0\tau}(x|x(0)))^\top | x(0)]] = 0, \tag{15}$$

*then we have:*

$$L_{SM} \leq L_{DSM} \quad and \quad W_2(p_0, q_0) \leq \sqrt{2 \left( \int_0^{T_s} g(\tau)^2 I(\tau)^2 \, d\tau \right) L_{DSM}} + I(T_s) W_2(p_{T_s}, q_{T_s}), \tag{16}$$

were $I(\tau) := \exp\{\int_0^\tau (L_f(r) + L_s(r)g(r)^2)dr\}$. Additionally, $L_f$ and $L_s$ are defined as follows:

(A1) The drift coefficient $f : \mathbb{R}^d \times [0, T_s] \to \mathbb{R}^d$ is Lipschitz continuous in the space variable $x$: there exists a positive constant $L_f(\tau) \in (0, \infty)$, depending on $\tau \in [0, T_s]$, such that for all $x, y \in \mathbb{R}^d$

$$\|f(x, \tau) - f(y, \tau)\| \leq L_f(\tau)\|x - y\|. \tag{17}$$

(A2) $s_\psi : \mathbb{R}^d \times [0, T_s] \to \mathbb{R}^d$ satisfies the one-sided Lipschitz condition [14, Definition 2.1]: there exists a constant $L_s(\tau) \in \mathbb{R}$, depending on $\tau \in [0, T_s]$, satisfying for all $x, y \in \mathbb{R}^d$

$$(s_\psi(x, \tau) - s_\psi(y, \tau)) \cdot (x - y) \leq L_s(\tau)\|x - y\|^2. \tag{18}$$

For more details, see Kwon et al. (2022). In our case $p_0 = p_n^b$ and $q_{T_s} = p_n^f$, the $n^{th}$ backward and forward marginals respectively. We assume the forward marginals are a "noisy" version of the backward marginals (Def. A.3) with a small $\sigma$. Thus, $p_{T_s} = q_{T_s}$ and $W_2(p_{T_s}, q_{T_s}) = 0$ the first time the score model is applied, (the starting point is on the forward marginal). Consequently, as the $L_{DSM}$ loss converges to zero, so is the bound on the Wasserstein distance.

**Conditions:** $Var[\mathbb{E}[(\nabla_x \log p_{0\tau}(x|x(0)))^\top | x(0)]] = 0$ under the following sufficient conditions: (1) Lipschitz continuity of the drift function for the forward diffusion process, and (2) boundedness of the noise schedule. Our scenario satisfies these conditions as follows:

- The drift function $f$ is zero, which is Lipschitz continuous.
- The noise schedule $g$ is between $\sigma_{min}$ and $\sigma_{max}$, thus it is bounded above and below.

Therfore, we meet these sufficient conditions and adopt the additional assumptions from Kwon et al. (2022), enabling us to apply Thm. A.4.

**Lemma A.5.** *Let $u : [0, 1] \times \mathbb{R}^d \to \mathbb{R}^d$ be a continuous time-dependent vector field, satisfying the Lipschitz condition: for any $t \in [0, 1]$ and $x, y \in \mathbb{R}^d$, $\|u_t(x) - u_t(y)\| \leq L\|x - y\|$. Let $p_n^b$ and $p_n^f$ represent the $n^{th}$ backward and forward marginals respectively, where the corresponding **marginals** continuous time is $t_n = n/N$. Let $s_\psi$ denote the trained score model. Then, applying $s_\psi$ from $p_n^f$ to $p_n^b$ as described in Sec. 2.2 lowers the final bound on $W_2(p_N^b, p_N^f)$:*

$$W_2(p_N^b, p_N^f) \leq \left( \sqrt{2 \left( \int_0^{T_s} g(\tau)^2 I(\tau)^2 d\tau \right) L_{DSM}^\psi} \right) \exp\{L(1 - t_n)\}.$$

*Proof of Lemma 3.7.* Denote as $d_n = W_2(p_n^b, p_n^f)$. After applying the score models $s_\psi$:

$$\hat{d}_{n,\psi} = W_2(p_n^b, \hat{p}_n^b) \leq \sqrt{2 \left( \int_0^{T_s} g(\tau)^2 I(\tau)^2 d\tau \right) L_{DSM}^\psi} + I(T_s) W_2(p_{n,\sigma}^b, p_n^f), \quad (19)$$

where $\hat{p}_n^b$ is the score model's approximation and $0, T_s$ are the integration times of the score model and $T_s \to \infty$. According to Lemma. A.1:

$$W_2(p_N^b, p_N^f) \leq \hat{d}_{n,\psi} \exp\{L(1 - t_n)\}, \quad (20)$$

$$W_2(p_N^b, p_N^f) \leq W_2(p_n^b, p_n^f) \exp\{L(1 - t_n)\} = W_2(p_n^b, p_{n,\sigma}^b) \exp\{L(1 - t_n)\}, \quad (21)$$

where the second inequality is the bound without applying the correction step.

The final bound can be obtained by substituting the bound on $\hat{d}_{n,\psi}$:

$$W_2(p_N^b, p_N^f) \leq \left( \sqrt{2 \left( \int_0^{T_s} g(\tau)^2 I(\tau)^2 d\tau \right) L_{DSM}^\psi} + I(T_s) W_2(p_{n,\sigma}^b, p_n^f) \right) \exp\{L(1 - t_n)\}. \quad (22)$$

Assuming the score model is trained, $J_{DSM}^\psi$ is negligible: $\sqrt{2 \left( \int_0^{T_s} g(\tau)^2 I(\tau)^2 d\tau \right) L_{DSM}^\psi} < W_2(p_n^b, p_{n,\sigma}^b)$. Since the forward marginals are a noisy version of the backward marginals $W_2(p_n^f, p_{n,\sigma}^b) = 0$. In total, the final bound after applying a correction step is lower than the bound when applying no correction steps:

$$\left( \sqrt{2 \left( \int_0^{T_s} g(\tau)^2 I(\tau)^2 d\tau \right) L_{DSM}^\psi} \right) \exp\{L(1 - t_n)\} < W_2(p_n^b, p_{n,\sigma}^b) \exp\{L(1 - t_n)\}. \quad (23)$$

$\square$

**Theorem A.6.** *Let $u : [0, 1] \times \mathbb{R}^d \to \mathbb{R}^d$ be a continuous time-dependent vector field, satisfying the Lipschitz condition: for any $t \in [0, 1]$ and $x, y \in \mathbb{R}^d$, $\|u_t(x) - u_t(y)\| \leq L \cdot \|x - y\|$. Denote $l^{\psi,n} = \sqrt{2 \left( \int_0^{T_s} g(\tau)^2 I(\tau)^2 d\tau \right) L_{DSM}^{\psi,n}}$, where $L_{DSM}^{\psi,n}$ refers to the loss of $s_\psi$ on the $n^{th}$ marginal. Following Lemma. 3.7 and assuming that $W_2(p_{n,\sigma_n}^b, q_{T_s,n}) \leq W_2(p_n^b, q_{T_s,n})$ for every $n$ where $q_{T_s,n}$ is the initial distribution of the $n^{th}$ correction of the score model, then the application of $s_\psi$ to multiple marginals upper bounds the final Wasserstein distance. This can be expressed as:*

$$W_2(p_N^b, p_N^f) \leq \sum_{i=0}^{N-1} l^{\psi,i} \exp\{(N - i) \cdot L/N\} I^{N-1-i}(T_s) + d_0 I^N(T_s) \exp\{L\}.$$

Under the assumption that the forward marginals are a noisy version of the backward marginals when $q_{T_s,n} = p_n^f$ the inequality: $W_2(p_{n,\sigma_n}^b, p_n^f) \leq W_2(p_n^b, p_n^f)$ is trivial. After applying a correction step and continuing the flow, the trajectory at the next time-step lies between the forward and backward marginal paths, rather than strictly on the forward marginals path ($q_{T_s,n} \neq p_n^f$). This inequality holds true provided that $\sigma_n$ decreases at a rate corresponding to this intermediate positioning.

**Corollary A.6.1.** *When no corrections are applied the final Wasserstein distance is:*

$$W_2(p_N^b, p_N^f) \leq d_0 \exp\{L\} \quad (24)$$

*Assuming the score model is trained $l^{\psi,i}$ is negligible. In our case the drift is $0$ resulting in a negligible $L_f(r)$. Additionally, the one-sided Lipschitz constant is $\lim_{\tau \to \infty} \sigma^2 L_s(\tau) = -1$ (Kwon et al., 2022). Therefore, $\exists \tau'$ such that $\forall \tau > \tau'$ $I(\tau) < 1$, specifically $\lim_{\tau \to \infty} I(\tau) < 1$. Applying more correction scores decreases the upper bound of the final Wasserstein distance by $I(T_s)$, similarly to the multiplicative case (B) in Thm. A.2.*

*Proof of Theorem A.6.* Denote as $d_n = W_2(p_n^b, q_{T_s,n})$. We will prove by induction that:

$$d_n \leq \sum_{i=0}^{n-1} l^{\psi,i} \exp\{(n-i) \cdot L/N\} I^{n-i-1}(T_s) + d_0 I^n(T_s) \exp\{n \cdot L/N\}. \tag{25}$$

Base case ($n = 0$): Trivially true.

Inductive step: Assume the inequality holds for $n$. We'll prove it for $n + 1$.

By Lemma. 3.7, after applying the score function:

$$\hat{d}_{n,\psi} = W_2(p_n^b, \hat{p}_n^b) \leq \sqrt{2 \left( \int_0^{T_s} g(\tau)^2 I(\tau)^2 d\tau \right) L_{DSM}^{\psi,n} + I(T_s) W_2(p_{n,\sigma_n}^b, q_{T_s,n})} \tag{26}$$

$$\leq \sqrt{2 \left( \int_0^{T_s} g(\tau)^2 I(\tau)^2 d\tau \right) L_{DSM}^{\psi,n} + I(T_s) W_2(p_n^b, q_{T_s,n})} \tag{27}$$

$$= \sqrt{2 \left( \int_0^{T_s} g(\tau)^2 I(\tau)^2 d\tau \right) L_{DSM}^{\psi,n} + I(T_s) d_n}, \tag{28}$$

in the second inequality we used the assumption that $W_2(p_{n,\sigma_n}^b, q_{T_s,n}) \leq W_2(p_n^b, q_{T_s,n})$. By Lemma. A.1 over the interval $[t_n, t_{n+1}]$, we have:

$$d_{n+1} \leq \hat{d}_{n,\psi} \exp\{L(t_{n+1} - t_n)\} \tag{29}$$

$$\leq \left( \sqrt{2 \left( \int_0^{T_s} g(\tau)^2 I(\tau)^2 d\tau \right) L_{DSM}^{\psi,n} + I(T_s) d_n} \right) \exp\{L(t_{n+1} - t_n)\} \tag{30}$$

$$= \left( \sqrt{2 \left( \int_0^{T_s} g(\tau)^2 I(\tau)^2 d\tau \right) L_{DSM}^{\psi,n}} \right) \exp\{L/N\} + d_n I(T_s) \exp\{L/N\}, \tag{31}$$

Substituting the inductive hypothesis:

$$d_{n+1} \leq l^{\psi,n} \exp\{L/N\} + d_n I(T_s) \exp\{L/N\} \tag{32}$$

$$\leq l^{\psi,n} \exp\{L/N\} \tag{33}$$

$$+ \left( \sum_{i=0}^{n-1} l^{\psi,i} \exp\{(n-i) \cdot L/N\} I^{n-i-1}(T_s) + d_0 I^n(T_s) \exp\{n \cdot L/N\} \right) I(T_s) \exp\{L/N\} \tag{34}$$

$$= \sum_{i=0}^{n} l^{\psi,i} \exp\{(n-i+1) \cdot L/N\} I^{n-i}(T_s) + d_0 I^{n+1}(T_s) \exp\{(n+1) \cdot L/N\}. \tag{35}$$

This completes the induction. The final bound at $t = 1$ follows by setting $n = N$.

$\square$

## A.3   PRACTICAL CONSIDERATIONS FOR $h_\psi$

**Practical Considerations of $h_\psi$:** When designing the learning objective for $h_\psi$ an important issue should be taken into consideration. The pre-trained flow model has established a matching between each distribution on the forward trajectory $p_{t_n}^f$ and between each distribution on the backward trajectory $p_{t_n}^b$ across different time-steps. Additionally, there exists a matching between $\hat{\pi}_1$ and $\pi_1$.

Through transitivity, this implies a pairing between $p_{t_n}^f$ and $p_{t_n}^b$ for every time step $t_n$. During the training process, samples from $p_{t_n}^f$ and $p_{t_n}^b$ are accessible. Models that try to establish pairing between these distributions will yield sub-optimal results, since the correct pairing is unknown. Thus, we opt for different generative models such as score and classifier.

**Alternative Correction Models:** Alternative correction models include the more sophisticated score models for traversing between forward and backward marginals, provided the noise level for the forward marginal is known. A supplementary model can be trained to predict the noise level, thereby assisting the score model. Additionally, Energy-Based Models (EBMs) (Du et al., 2021), could also be a good fit to explore the alignment of the sampling trajectory with the backward trajectory.

### A.3.1 PARALLEL SAMPLING

Fig. 6 presents the approximation of parallel sampling the correction and flow model. The algorithm for parallel sampling is presented in Alg. 3. For clarity, the algorithm assumes Euler integration as the ODE-solver. To maintain computational efficiency, a single correction step is implemented, ensuring that the total computation time remains equivalent to that of the uncorrected method.

This algorithm is parallel in principle. Cuda has a queue, where the CPU sends tasks to be run on the GPU. The GPU may execute tasks in parallel that are independent of one another, such as our parallel sampling. The CPU may wait when the queue is full or during synchronization events, (i.e. item(), synchronize(), etc). Given a powerful enough Cuda machine, the models could be run in parallel on different GPUs or using advanced parallelism techniques. Parallel calculation on separate devices could be useful when the correction calculation takes more time than the overhead of moving between devices, as is the case for large models. The extent of parallel processing viability will depend on the specific hardware and infrastructure available.

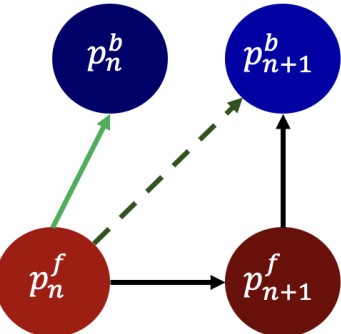

Figure 6: **Parallel Sampling** The black arrows denote the path of first flowing with $v_\theta$ and only then taking a correction step with $h_\psi$. In contrast, the dashed green arrow shows the approximation obtained by summing the direction of the flow model $v_\theta$ (black arrow), and the correction model $h_\psi$ (green arrow) on the current step.

---

**Algorithm 3** Parallel Corrected Inference

---

**Require:** flow model $v_\theta$, correction model $h_\psi$, number of iterations $N$, step size of corrector steps $\{\alpha_i\}_{i=0}^{N-1}$, scale added noise $\{\beta_i\}_{i=0}^{N-1}$
1: $x_0 \sim \mathcal{N}(0, I)$
2: **for** $n = 0, \ldots, N-1$ **do**:
3: $\quad \epsilon \sim \mathcal{N}(0, I)$
4: $\quad \tilde{x}_{t_n} = x_{t_n} + \beta_n \cdot \epsilon$
5: $\quad h_{t_n} = h_\psi(t_n, \tilde{x}_{t_n})$ $\hfill \triangleright$ Correction
6: $\quad v_{t_n} = v_\theta(t_n, x_{t_n})$ $\hfill \triangleright$ Flow
7: $\quad x_{t_{n+1}} = x_{t_n} + \frac{1}{N} v_{t_n} + \alpha_n \cdot h_{t_n}$
8: **end for**
9: **return** $x_{t_N}$

---

### A.3.2 SCORE MODEL TRAINING

Score models are generative AI systems that transform random noise into meaningful data through iterative denoising. At their core is the score function - the gradient of the log probability density - which guides samples toward higher likelihood regions of a target distribution.

These models utilize varying noise levels ($\sigma$) during generation. At high noise levels, samples differ significantly from the target distribution, but the score function provides stable gradients far from the data manifold. At low noise levels, as samples approach the target distribution, the score function enables detailed refinement. During generation, the model progressively reduces noise levels while following each corresponding score function, establishing a path between random noise and complex data distributions.

By assuming the forward marginals are a noisy version of the backward marginals (with a small $\sigma$), a score model can be used to transverse between them.

While sophisticated approaches like annealing score models with time-varying $\sigma$ are widely used for generation, they proved unsuitable for our needs. This is due to our inference process that begins at a forward marginal (or an intermediate point), where we lack information about the current noise level, precluding the effective use of such models.

### A.3.3 ROBUST CLASSIFIER TRAINING

**Adversarial Training (AT):** Srinivas & Fleuret (2021) demonstrated that the gradients of standard classifiers can be arbitrarily altered without impacting their cross-entropy loss or accuracy. Building on this insight, Kawar et al. (2023) proposed using gradients from a *robust* classifier for guidance, rather than those from a conventional classifier. The gradients of robust classifiers are resistant to arbitrary manipulation as a results of the adversarial attack used in their loss computation is directly dependent on the model's gradients. An intriguing characteristic of robust classifiers is that their gradients have been shown to align well with human perception, as noted by Tsipras et al. (2019). When robust classifiers are employed to guide $x$ towards a specific class c, they are anticipated to produce significant features that correspond well with the target class. As a result, the modifications applied to $x$ are likely to be visually convincing and aligned with human perception of the class characteristics. Inspired by these works we enhance the classifier with Projected Gradient Descent (PGD) attack and Adversarial Training (AT) robustification method (Madry et al. (2018)), with the PGD pseudo-algorithm detailed in Alg. 4. Our findings indicate that adversarial training stabilizes gradients and enhances their meaningfulness.

---

**Algorithm 4** Targeted Projected Gradient Descent

---

**Require:** classifier $f_\phi$, input $x$, class $c$, number of iterations $N$, step size $\alpha$, radius $\epsilon$, loss function $\ell$,
  1: $\delta_0 = 0$
  2: **for** $n = 0, \ldots, N$ **do**:
  3:   $\delta_{n+1} = \Pi_\epsilon(\delta_n - \alpha \nabla_\delta \ell(f_\phi(x + \delta_t), c))$
  4: **end for**
  5: $x_{ADV} = x + \delta_N$
  6: Return $x_{ADV}$

---

Where $\Pi_\epsilon$ is a projection operator that keeps $\delta$'s norm below $\epsilon$ so the adversarial example will not stray too far from the input; and the loss function in our case is cross entropy.

**Gradient Alignment (GA):** Inspired by Yadin et al. (2024) and Song et al. (2021b), we incorporate a cosine-similarity loss term for the gradient. This aims to align the classifier's gradient direction with the backward marginals when in close proximity. We artificially move away from the backward marginals by introducing small Gaussian perturbations to sampled points, and training the classifier gradient to point towards the original samples by aligning them with the negative direction of the noise.

---

**Algorithm 5** Robust Classifier Training

---

**Require:** classifier $h_\psi$, forwards marginals $\{p_{t_n}(x)^f\}_{n=0}^N$, backward marginals $\{p_{t_n}(x)^b\}_{n=0}^N$, loss weights $\{\beta_i\}_{i=1}^3$, $\sigma$ scale of noise, adversarial step size $\alpha$, adversarial number of steps $K$, adversarial radius $\epsilon_{ADV}$
  1: **repeat**
  2:   $\epsilon \sim \mathcal{N}(0, I)$
  3:   $n \sim U(\{0, 1, .., N\})$
  4:   $s = \mathcal{U}(0, 1) \cdot \sigma$
  5:   $x_f \sim p_{t_n}^f, x_b \sim p_{t_n}^b$
  6:   $\ell_{CE} = CE(h_\psi(x_f, t_n), 0) + CE(h_\psi(x_b, t_n), 1)$
  7:   $c_n = \nabla_\psi h_\psi(x_b + s \cdot \epsilon, t_n)$
  8:   $\ell_{CS} = cosine - similarity(c_n, -\epsilon)$
  9:   $\ell_{ADV} = PGD(h_\psi, x_f, 0, K, \alpha, \epsilon_{ADV}, CE) + PGD(h_\psi, x_b, 1, K, \alpha, \epsilon_{ADV}, CE)$
  10:   $\ell = \beta_1 \cdot \ell_{CE} + \beta_2 \cdot \ell_{CS} + \beta_3 \cdot \ell_{ADV}$
  11:   Take gradient descent step on $\nabla_\psi \ell$
  12: **until** converged

---

The class for the forward marginals is represented by $0$ and for the backward marginals by $1$.

## A.4 EXTENDED RELATED WORK

**Inverse diffusion** – Previous work researched inversion of diffusion models, which are "approximately invertible" models, particularly in the context of image editing. DDIM inversion (Song et al., 2021a) method laid the groundwork for this approach, enabling the reversal of the diffusion process to obtain latent representations of images by utilizing a deterministic forward process. Building upon this foundation, Wallace et al. (2023) improved the efficiency of the inversion by training an encoder network to directly map images to their corresponding noise representations in the diffusion process. Furthermore, Mokady et al. (2023) enabled precise edits on real images through a text-guided approach, using a null-text optimization to find an optimal noise that, when denoised, produces the target image. Zhang & Kleijn (2023) improved the accuracy of the inversion process by combining forward and backward trajectories to minimize approximation errors, while Pan et al. (2023) focused on enhancing both the speed and quality of image editing operations by iteratively refining the inverted latent code.

## A.5 FORWARD AND BACKWARD TRAJECTORY COMPARISON

We illustrate the distinction between the forward and backward marginals of OT-CFM flow model (Tong et al., 2024) trained on the CIFAR-10 dataset and sampled with 10-step RK4. Fig. 9 presents a visual representation of these marginals at each time-step $t_n$, created with t-SNE (Van der Maaten & Hinton (2008)) in order to reduce the marginals' dimensionality to two. Fig. 8 offers an alternative visualization, where the t-SNE is applied to the classifier correction model's features of the marginals at each time-step. The visualizations demonstrate that as $n$ increases there is a trend of growing similarity between the forward and backward marginals. However, this pattern of convergence does not extend to the final time step ($n = 10$), where the comparison is between the actual data and the flow model's approximation of it.

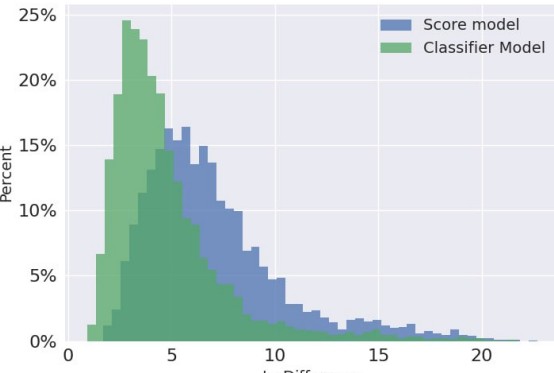

Figure 7: Percentage vs $L_2$ difference. Histogram showing $L_2$ differences of VGG features between samples generated using a flow model alone versus with a correction model. The samples are generated from identical Gaussian noise. The percentage indicates the proportion of images with the corresponding difference. The score and classifier correction models leave most images largely unchanged, with differences concentrated in a small range, while significantly altering only outliers. This suggests that there is an intersection between the forward and backward marginals.

Reinforcing this observation, Fig. 7 examines the intersection of forward and backward marginals by presenting a histogram of the $L_2$ differences in VGG[5] features between uncorrected and corrected samples. The distribution reveals that for both the classifier and score models, the majority of images exhibit minimal changes. This pattern indicates that corrections are primarily applied to images requiring adjustment, suggesting a close alignment between the forward and backward marginals. The selective nature of these corrections implies that the models effectively identify and address outliers, refining the overall distribution while leaving well-formed samples largely unchanged.

---

[5]VGG is the model used to calculate the FID score

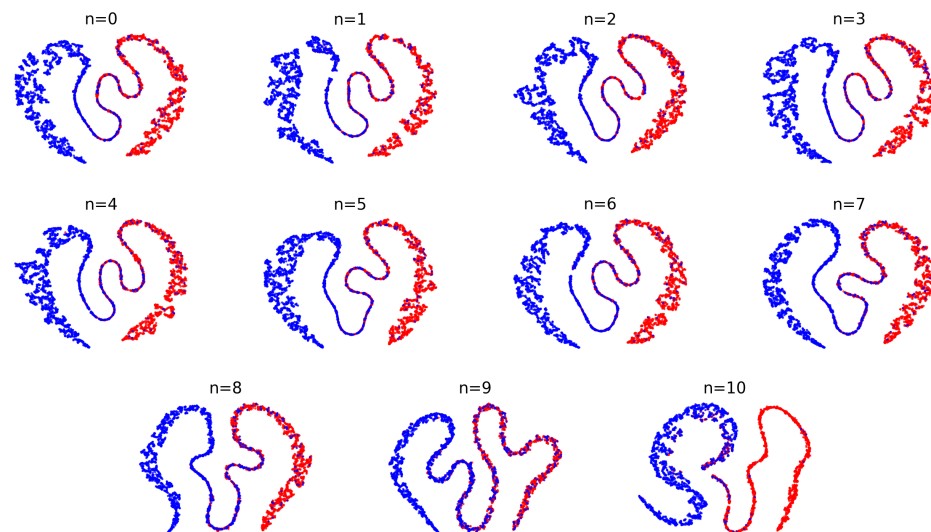

Figure 8: **TSNE of Classifier Features** Red represents the forward marginals, while blue denotes the backward marginals. The classifier's features can differentiate between the forward and backward marginals in most cases. However, in some instances, these marginals are indistinguishable from one another.

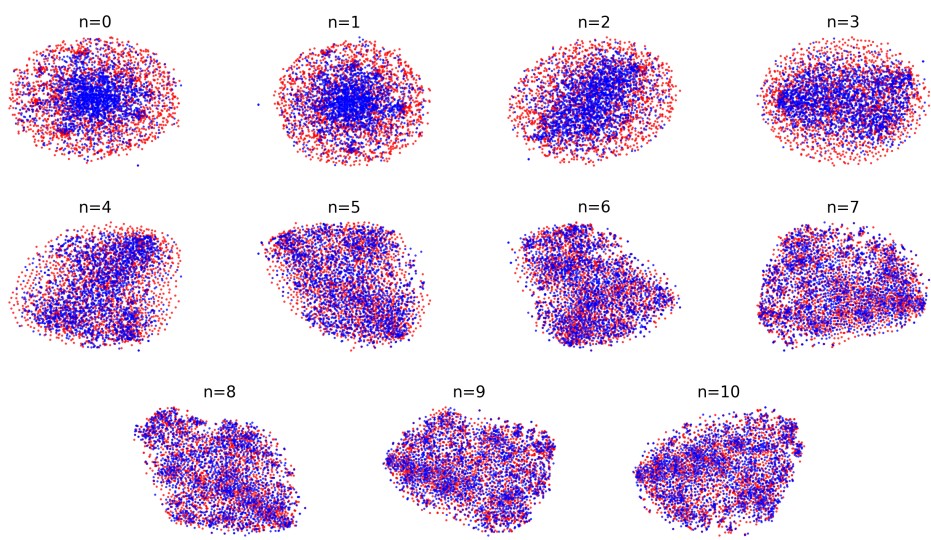

Figure 9: **TSNE of Forward and Backward Marginals** Red represents the forward marginals, while blue denotes the backward marginals. As time progresses, we observe a convergence between these two sets of marginals. However, it's important to note that despite this increasing proximity, they never achieve perfect alignment or identity. $n = 0$ compares the approximate source distribution distribution ($p_0^b$) and Gaussian noise while $n = 10$ compares the model's data approximation ($p_1^f$) and the real data.

## A.6 Additional Experiments

### A.6.1 Score Ablation Study

An ablation study on different $\sigma$ values for the score model trained on CIFAR-10 is presented in Table. 3. The correction step-sizes for different sigmas were multiplied by a constant value, as the

score function is multiplied by sigma during evaluation. For all other evaluations, we employed the score model trained using $\sigma = 0.005$, as it yielded superior performance compared to other values.

| Correction Model | NFE ↓ | C-NFE ↓ | FID ↓ |
|---|---|---|---|
| 0.001-Score | 41 | 1 | 3.83 |
| | 43 | 3 | **3.75** |
| | 51 | 11 | 3.77 |
| 0.003-Score | 41 | 1 | 3.51 |
| | 43 | 3 | **3.44** |
| | 51 | 11 | 3.45 |
| 0.005-Score | 41 | 1 | 3.45 |
| | 43 | 3 | 3.38 |
| | 51 | 11 | **3.37** |
| 0.01-Score | 41 | 1 | 3.49 |
| | 43 | 3 | 3.42 |
| | 51 | 11 | **3.38** |

Table 3: Ablation study on the sigma value of score correction model sampled on $n = \{[10], [0, 5, 10], \forall n\}$ marginals on CIFAR-10. The $\sigma-$Score represents the value of $\sigma$ the score model was trained with. The base flow model is OT-CFM sampled with 10-step RK4 (40 NFE). The results demonstrate that the optimal performance is achieved when $\sigma = 0.005$ (the value used in the main paper).

### A.6.2 CIFAR-10 CLASSIFIER GUIDANCE

We implement classifier guidance as described in Sec. 3.2.2 on CIFAR-10 dataset. Fig. 10 presents images produced with our corrected inference algorithm that use the class-condition classifier, where the trajectory is steered toward the correct class in the backward marginals. On the left, the original images of the flow model are presented (with no correction), and on the right the images from the same source noise, but with correction steps toward the matching classes. Even when the classifier's class matches the original class of the source noise it produces a different image of the same class, more closely aligned with the backward trajectory.

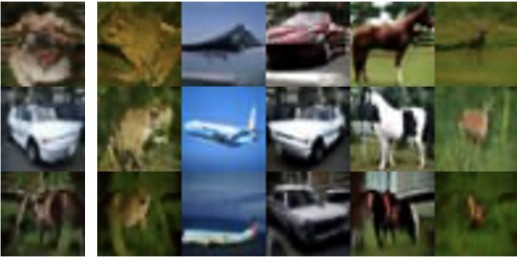

Figure 10: CIFAR-10 images generated by OT-CFM sampled with 10-step RK4, shown alongside their counterparts produced with class conditioned classifier correction. The classifier successfully directs the noise to the correct class.

### A.6.3 ADDITIONAL CIFAR-10 SAMPLE QUALITY RESULTS

Table. 4 presents the complete results on CIFAR-10 of applying the correction models (score and classifier) in parallel with the flow model, (to save computation time), and with an approximation of the backward marginals (to save storage during training). For more details see Sec. 5.1.

| Correction Model | NFE ↓ | C-NFE ↓ | OT-CFM FID ↓ | I-CFM FID ↓ |
|---|---|---|---|---|
| No Correction | 40 | 0 | 4.34 | 4.29 |
| Classifier | 42 | 1  ($n = 8$) | 3.57 | 3.77 |
|  | 46 | 3  ($n = 8, 9, 10$) | 3.48 | 3.67 |
|  | 50 | 5  ($n = 6, 7, 8, 9, 10$) | **3.47** | **3.62** |
|  | 62 | 11 ($\forall n$) | 3.48 | 3.63 |
| P-Classifier | 40 | 1  ($n = 7$) | 3.65 | 3.75 |
|  | 40 | 3  ($n = 7, 8, 9$) | 3.60 | 3.73 |
|  | 40 | 5  ($n = 5, 6, 7, 8, 9$) | **3.59** | **3.67** |
|  | 40 | 10 ($n = 0, 1, 2, 3, 4, 5, 6, 7, 8, 9$) | 3.74 | 3.80 |
| Classifier (Backward Marginals Interpolation) | 42 | 1  ($n = 8$) | 3.77 | 3.74 |
|  | 46 | 3  ($n = 8, 9, 10$) | 3.72 | **3.62** |
|  | 50 | 5  ($n = 6, 7, 8, 9, 10$) | **3.67** | 3.69 |
|  | 62 | 11 ($\forall n$) | 3.68 | 3.67 |
| Score | 41 | 1  ($n = 10$) | 3.45 | 3.47 |
|  | 43 | 3  ($n = 0, 5, 10$) | 3.38 | 3.39 |
|  | 51 | 11 ($\forall n$) | **3.37** | **3.38** |
| P-Score | 40 | 1  ($n = 9$) | 4.29 | 4.27 |
|  | 40 | 3  ($n = 0, 5, 9$) | **4.08** | **3.99** |
|  | 40 | 10 ($n = 0, 1, 2, 3, 4, 5, 6, 7, 8, 9$) | **4.08** | **3.99** |
| Score (Backward Marginals Interpolation) | 41 | 1  ($n = 10$) | 3.46 | 3.47 |
|  | 43 | 3  ($n = 0, 5, 10$) | **3.38** | **3.40** |
|  | 51 | 11 ($\forall n$) | 3.41 | 3.41 |

Table 4: Comparison of two variants of classifier and score correction models on CIFAR-10. First, a correction model that runs in parallel with ("P-") the flow model. Second, a correction model that learns an interpolation of the data and approximate source distribution. The base flow models are sampled with 10-step RK4 (40 NFE). The classifier succeeds in improving the FID even with the parallel approximation, while the score shows less improvement. Training with interpolated backward marginals improves the FID score, though less than using their exact version.

A.6.4   FLOW AS CORRECTION MODEL

In this experiment we perform MM with a flow model (OT-CFM) as the correction model. The correction is performed only on the first and last marginals with a different number of correction steps, see Table. 5. The correction of the first marginal is done between Gaussian noise ($p_0^f$) to the approximate source distribution ($p_0^b$), and on the last marginal from the approximate target distribution ($p_1^f$) to the target distribution ($p_1^b = \pi_1$). The first row in the table presents the FID result with no correction model, but only 10-step RK4 sampling of the OT-CFM base flow model.

The flow model only degrades the results. We hypothesize that this is due to the pairing problem, for more details refer to Appendix. A.3. That led us to explore other correction models - score and robust classifier.

| Correction Model | NFE↓ | C-NFE↓ | FID ↓ |
|---|---|---|---|
| No Correction | 40 | 0 | 4.34 |
| OT-CFM – $1^{st}$ Marginal | 42 | 2 | 9.74 |
|  | 46 | 6 | 6.31 |
|  | 60 | 20 | 6.33 |
| OT-CFM – $N^{th}$ Marginal | 42 | 2 | 7.43 |
|  | 46 | 6 | 4.39 |
|  | 60 | 20 | 4.41 |

Table 5: CIFAR-10 FID comparison of OT-CFM correction and flow model trained on first and last marginals. The correction score model was sampled with C-NFE RK4 steps, while the base flow model was sampled with 10-step RK4 (40 NFE). The correction flow model degrades the results.

### A.6.5 CIFAR-10 TEST SAMPLE QUALITY

**CIFAR-10 Test Set FID:** The performance of OT-CFM flow model with and without correction on CIFAR-10 test set is presented in Table 6. The correction model helps the generation quality, even though it was trained to match the training data marginals. Where the C-NFE is greater than one, the correction steps were taken on different time-step marginals; we examine several options: for the score model the correction steps were taken on $n = [10], [0, 5, 10], \forall n$ and for the classifier on $n = [8], [8, 9, 10], [6, 7, 8, 9, 10], \forall n$, same as Table .1.

| Flow Model | NFE ↓ | FID ↓ |
|---|---|---|
| OT-CFM | 40 | 6.43 |
| | 80 | 5.51 |
| | 200 | 5.67 |

| Correction Model | NFE ↓ | C-NFE ↓ | | FID ↓ |
|---|---|---|---|---|
| Score | 41 | 1 | $(n = 10)$ | 5.52 |
| | 43 | 3 | $(n = 0, 5, 10)$ | 5.46 |
| Classifier | 42 | 1 | $(n = 8)$ | 5.66 |
| | 46 | 3 | $(n = 8, 9, 10)$ | 5.57 |
| | 50 | 5 | $(n = 6, 7, 8, 9, 10)$ | 5.56 |

Table 6: CIFAR-10 test set FID performance of OT-CFM flow model with and without our correction models (score and classifier). OT-CFM is sampled without correction using 10,20, and 50-step RK4. Our correction models use OT-CFM with 10-step RK4 (40 NFE). In general, adding correction steps (C-NFE) improves the results.

### A.6.6 FLOW MODELS RK4 RESULTS

| Model | NFE | FID ↓ |
|---|---|---|
| OT-CFM (Tong et al., 2024) | 40 | 4.34 |
| | 44 | 3.96 |
| | 48 | 3.73 |
| | 52 | 3.59 |
| | 56 | 3.52 |
| | 60 | 3.47 |
| | 80 | 3.48 |
| | 160 | 3.65 |
| | 200 | 3.67 |
| | 400 | 3.69 |
| I-CFM (Tong et al., 2024) | 40 | 4.29 |
| | 44 | 3.96 |
| | 48 | 3.74 |
| | 52 | 3.60 |
| | 56 | 3.52 |
| | 60 | 3.47 |
| | 80 | 3.47 |
| | 160 | 3.63 |
| | 200 | 3.64 |
| | 400 | 3.66 |

Table 7: CIFAR-10 FID scores for OT-CFM and I-CFM flow models sampled with RK4. The number of RK4 steps is $1/4$ of the number of the NFEs.

## A.7 IMAGENET-64 QUALITAIVE EXAMPLES

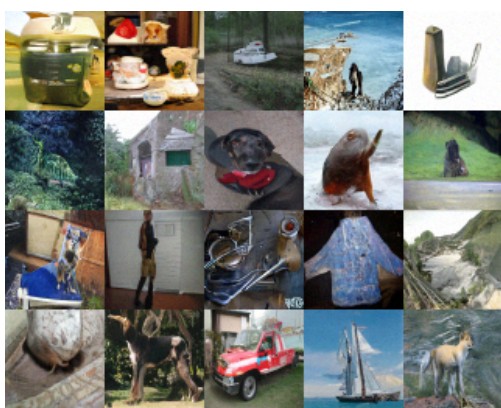

(a) OT-CFM

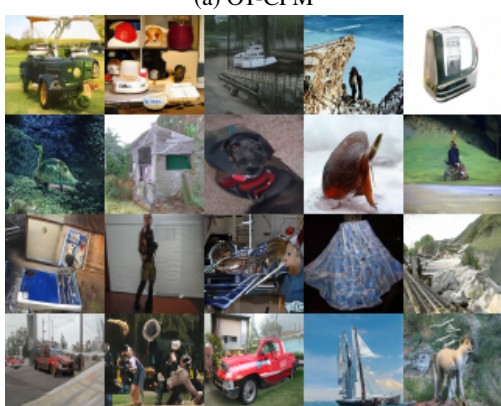

(b) Score Correction Model

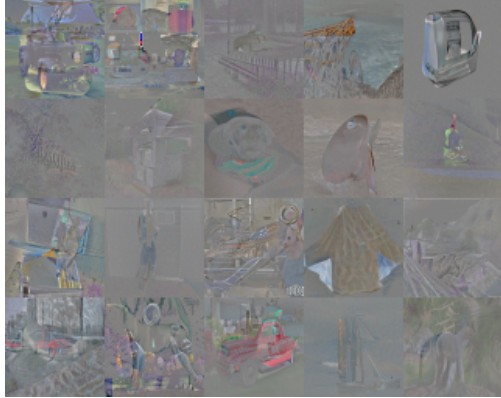

(c) Difference

Figure 11: ImageNet-64 image generation using OT-CFM alone and with score correction model. The corrected images demonstrate enhanced sharpness and definition compared to their uncorrected counterparts.

## A.8 CIFAR-10 QUALITAIVE EXAMPLES

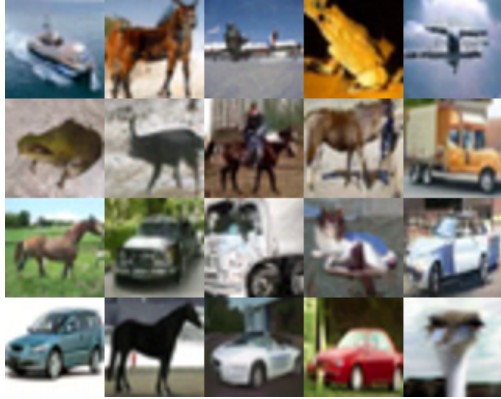

(a) OT-CFM

(b) Score Correction Model

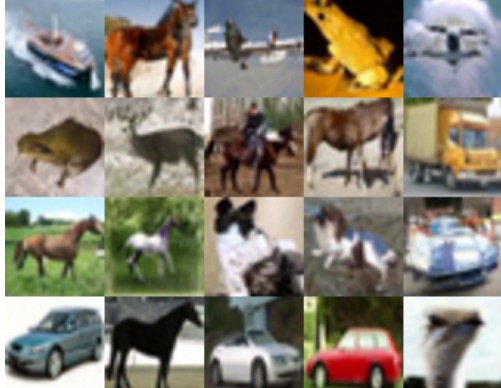

(c) Classifier Correction Model

Figure 12: CIFAR-10 image generation using OT-CFM alone and with score correction and classifier correction models. Despite the distinct nature of these correction models, their suggested improvements often exhibit remarkable similarity. The corrected images demonstrate enhanced sharpness and definition compared to their uncorrected counterparts.

### A.9 LAST STEP SCORE CORRECTION EXAMPLES

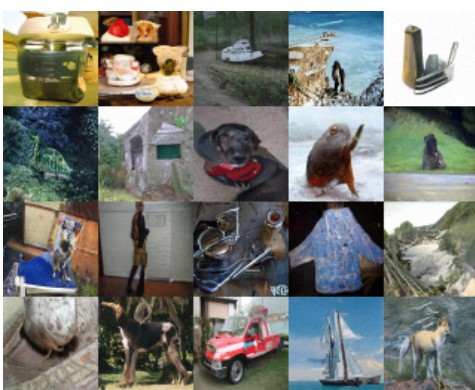

(a) OT-CFM

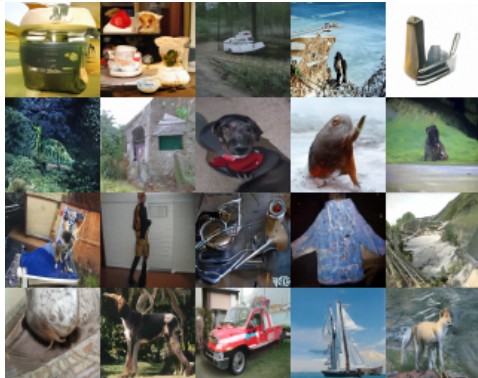

(b) Last Marginal Score Correction Model

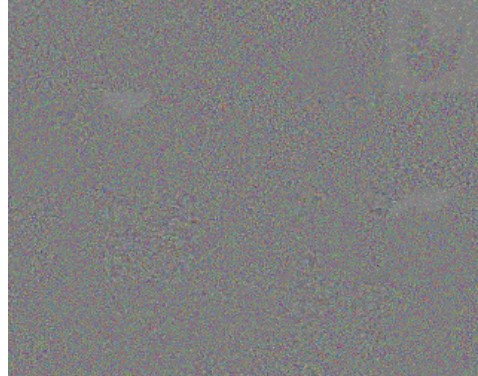

(c) Correction Difference

Figure 13: ImageNet-64 images generated by OT-CFM alone and with score correction model applied solely to the final step, accompanied by their difference (amplified for visibility). The noise-like appearance of the difference suggests the presence of residual noise in the model's predictions.

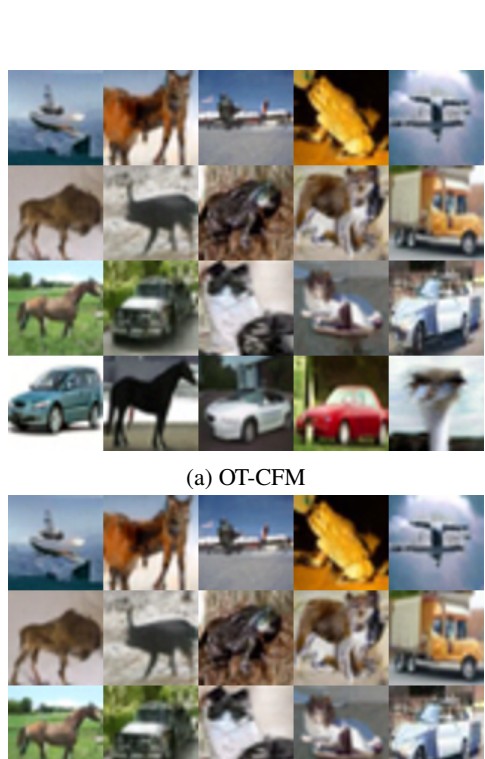

(a) OT-CFM

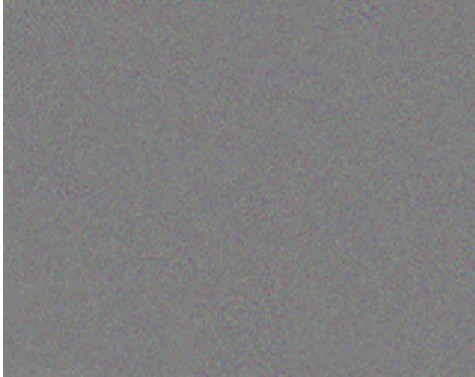

(b) Last Marginal Score Correction Model

(c) Correction Difference

Figure 14: CIFAR-10 images generated by OT-CFM alone and with score correction model applied solely to the final step, accompanied by their difference (amplified for visibility). The noise-like appearance of the difference suggests the presence of residual noise in the model's predictions.

## A.10    IMPLEMENTATION DETAILS

### A.10.1    FLOW MODELS ARCHITECTURE

Our flow models architecture is based on the UNet design from Nichol & Dhariwal (2021) that was employed in Tong et al. (2024) and Lipman et al. (2023).

|  | CIFAR-10 | ImageNet-64 |
|---|---|---|
| Channels | 128 | 192 |
| Depth | 2 | 3 |
| Channels multiple | 1,2,2,2 | 1,2,3,4 |
| Heads | 4 | 4 |
| Heads Channels | 64 | 64 |
| Attention resolution | 16 | 32,16,8 |
| Dropout | 0.1 | 0.0 |
| Batch size | 128 | 512 |
| Iterations | 400K | 600k |
| Learning Rate | 5e-4 | 1e-4 |
| Learning Rate Scheduler | Polynomial Decay | Constant |
| Warmup Steps | 5000 | 0 |

Table 8: Hyper-parameters used for CIFAR-10 and ImageNet-64 flow models.

### A.10.2 CORRECTIONS MODELS ARCHITECTURE

The correction models utilized the same UNet architecture with different hyper-parameters than the flow models. The classifier model required an additional convolution layer and two linear layers to reduce the output dimension to a single scalar. The correction models were trained for $100,000$ optimization steps, however we observed convergence within $30,000$ to $40,000$ steps. The models sizes are **half** the number of trainable parameters of the original flow models, requiring **significantly** less time to converge.

|  | Classifier | Score | |
|---|---|---|---|
|  | CIFAR-10 | CIFAR-10 | ImageNet-64 |
| Channels | 128 | 128 | 192 |
| Depth | 1 | 1 | 2 |
| Channels multiple | 1,2,1,2 | 1, 2, 1, 2 | 1, 2, 3, 2 |
| Heads | 4 | 4 | 4 |
| Heads Channels | 32 | 32 | 64 |
| Attention resolution | 16 | 16 | 32, 16, 8 |
| Dropout | 0.1 | 0.1 | 0.0 |
| Batch size | 128 | 128 | 512 |
| Iterations | 100K | 100K | 100K |
| Learning Rate | 0.0001 | 0.0002 | 0.0001 |
| $\sigma$ | 0.01 | 0.005 | 0.05 |
| Linear Layers Output | 64, 1 | - | - |

Table 9: Correction models hyper-parameters.

### A.10.3 LOSSES

**Score Model**: Our score model was trained using the denoising score matching loss (DSM) over the backward marginals $\{p_{t_n}^b\}_{n=0}^N$ with a constant noise scale $\sigma$:

$$L_{DSM} = \mathbb{E}\left[\mathbb{E}_{y_{t_n} \sim p_{t_n}^b, \epsilon \sim \mathcal{N}(0,\sigma^2 I)}\left[\|s_\psi(t_n, y_{t_n} + \epsilon) + \epsilon\|_2^2\right]\right]$$

where $s_\psi(t_n, y_{t_n})$ is the score model parameterized by $\psi$. For more details see Sec. 2.2.

**Classifier Model**: Our robust classifier was trained using $L_{\text{classifier}}$ which is comprised of 3 terms:

$$L_{\text{classifier}} = L_{BCE} + L_{AT} + L_{GA}$$

**Binary Cross-Entropy**: The binary cross-entropy (BCE) is used to distinguish between the backward marginals $p_{t_n}^b$ and the forward marginals $p_{t_n}^f$:

$$L_{BCE} = \mathbb{E}\left[\mathbb{E}_{y_{t_n} \sim p_{t_n}^b, x_{t_n} \sim p_{t_n}^f}\left[BCE(c_\psi(t_n, y_{t_n}), 1) + BCE(c_\psi(t_n, x_{t_n}), 0)\right]\right]$$

where $c_\psi(t_n, y)$ is the classifier output for input $y$ at time $t_n$, parameterized by $\psi$.

**Adversarial Training (AT)**: To enhance its robustness, the classifier was also trained on adversarial examples. This process involves:

1. Sampling $y_{t_n} \sim p_{t_n}^b$ and $x_{t_n} \sim p_{t_n}^f$.
2. Optimizing norm-clipped additive perturbations $\eta_y$ and $\eta_x$ to:
   - Decrease the classifier value for $y$: $c_\psi(t_n, y_{t_n} + \eta_y)$
   - Increase the classifier value for $x$: $c_\psi(t_n, x_{t_n} + \eta_x)$

For a more detailed explanation see Sec. A.3.3.

$$L_{AT} = \mathbb{E}\left[\mathbb{E}_{y_{t_n} \sim p_{t_n}^b, x_{t_n} \sim p_{t_n}^f}\left[BCE(c_\psi(t_n, y_{t_n} + \eta_y), 1) + BCE(c_\psi(t_n, x_{t_n} + \eta_x), 0)\right]\right]$$

**Gradient Alignment (GA)**: To align the classifier's gradient near the backward marginals, we introduce a small amount of Gaussian noise and use cosine similarity to adjust the classifier's gradient in the direction opposite to the noise (towards the backward marginal).

$$L_{GA} = \mathbb{E}\left[\mathbb{E}_{y_{t_n} \sim p_{t_n}^b, \epsilon \sim \mathcal{N}(0, \sigma^2 I)}\left[1 - \frac{\langle \nabla_{y_{t_n}} c_\psi(t_n, x_{t_n} + \epsilon), \epsilon \rangle}{\|\nabla_{y_{t_n}} c_\psi(t_n, x_{t_n} + \epsilon)\|_2 \|\epsilon\|_2}\right]\right]$$

### A.10.4 EVALUATION PARAMETERS

**Hyper-parameters:**

*Step size $\alpha$:* The configuration with the best FID was selected from a grid search over the interval $[0, 2]$ with a step size of 0.05. For the classifier's final step, the grid search was conducted over the interval $[0, 0.1]$ with a step size of 0.01.

*Noise $\beta$:* The configuration with the best FID was selected from a grid search over the interval $[0, 0.1]$ with a step size of 0.01.

**Sampling:** The hyper-parameters for all evaluations are described below, except for parallel sampling, where the time-steps are shifted by 1 ($10 \to 9, 9 \to 8$, etc). Additionally, for parallel classifier the final step-size is 0.02 instead of 0.06.

| Corr. Step | Classifier | | | | | | | |
|---|---|---|---|---|---|---|---|---|
| | 10 Steps | | 5 Steps | | 3 Steps | | 1 Step | |
| | Step Size | Noise | Step Size | Noise | Step Size | Noise | Step Size | Noise |
| 0 | 1.5 | 0.05 | - | - | - | - | - | - |
| 1 | 1.5 | 0.0 | - | - | - | - | - | - |
| 2 | 1.5 | 0.0 | - | - | - | - | - | - |
| 3 | 1.5 | 0.0 | - | - | - | - | - | - |
| 4 | 1.5 | 0.0 | - | - | - | - | - | - |
| 5 | 1.5 | 0.0 | - | - | - | - | - | - |
| 6 | 1.5 | 0.08 | 1.5 | 0.08 | - | - | - | - |
| 7 | 1.5 | 0.0 | 1.5 | 0.0 | - | - | - | - |
| 8 | 1.5 | 0.0 | 1.5 | 0.0 | 1.0 | 0.05 | 1.0 | 0.05 |
| 9 | 1.5 | 0.0 | 1.5 | 0.0 | 0.4 | 0.0 | - | - |
| 10 | 0.06 | 0.0 | 0.06 | 0.0 | 0.06 | 0.0 | - | - |

Table 10: CIFAR-10 evaluation hyper-parameters for the classifier correction model.

| Corr. Step | Score | | | | | |
|---|---|---|---|---|---|---|
| | 10 Steps | | 3 Steps | | 1 Step | |
| | Step Size | Noise | Step Size | Noise | Step Size | Noise |
| 0 | 0.4 | 0.03 | 0.35 | 0.03 | - | - |
| 1 | 0.3 | 0.0 | - | - | - | - |
| 2 | 0.3 | 0.0 | - | - | - | - |
| 3 | 0.3 | 0.0 | - | - | - | - |
| 4 | 0.3 | 0.01 | - | - | - | - |
| 5 | 0.35 | 0.05 | 0.45 | 0.05 | - | - |
| 6 | 0.3 | 0.0 | - | - | - | - |
| 7 | 0.3 | 0.0 | - | - | - | - |
| 8 | 0.3 | 0.01 | - | - | - | - |
| 9 | 0.3 | 0.00 | - | - | - | - |
| 10 | 2.0 | 0.0 | 2.0 | 0.0 | 2.0 | 0.0 |

Table 11: CIFAR-10 evaluation hyper-parameters for score correction model.

| Corr. Step | Score | | | | | |
|---|---|---|---|---|---|---|
| | 5 Steps | | 2 Steps | | 1 Step | |
| | Step Size | Noise | Step Size | Noise | Step Size | Noise |
| 0 | 0.4 | 0.1 | 0.4 | 0.01 | - | - |
| 1 | 0.2 | 0.05 | - | - | - | - |
| 2 | 0.2 | 0.0 | - | - | - | - |
| 3 | 0.2 | 0.0 | - | - | - | - |
| 4 | 0.2 | 0.05 | - | - | - | - |
| 5 | 0.4 | 0.0 | 0.4 | 0.0 | 0.4 | 0.0 |

Table 12: ImageNet-64 evaluation hyper-parameters for score correction model.

