# OpenReview forum: "Correcting Flows with Marginal Matching"
_ICLR.cc/2025/Conference — Submitted to ICLR 2025_

### Official Review · Reviewer_k9VA · 2024-10-28

**Soundness:** 3
**Presentation:** 2
**Contribution:** 3
**Rating:** 5
**Confidence:** 3

**Summary:**

This paper proposes a mitigation strategy for prediction errors in flow-based generative models. While previous research has often focused on minimizing discretization errors (by optimizing the mapping between Gaussian noise and the target data distribution), this work is the first to specifically target the prediction errors in flow-matching models.

The authors propose leveraging time reversibility in flow matching models. By implementing a corrective mechanism that modifies the generated sample trajectories during inference, the model can reduce errors between the generated and target distributions. This correction, applied intermittently with the ODE solver steps, is theoretically shown to minimize the Wasserstein distance between the generated and target distributions, under certain assumptions.

The paper presents two implementations of this correction mechanism: one that uses a score model and one that is based on a classifier. Empirical results show that this method improves generation quality on CIFAR-10 and ImageNet-64, achieving better FID scores with fewer sampling steps, especially in the former. The contributions of the paper include a novel time-reversal framework, theoretical analysis, and practical implementations to enhance flow-matching model performance.

**Strengths:**

- The paper presents an approach to improve flow models by enforcing that the forward and backward marginals need to match. This is a sensible thing to try.

**Weaknesses:**

The presentation of the paper can be improved. Some specific points that need to be changed:
- The statement of Theorem 3.3 is too long. It should be broken up.
- The explanation on the score model approach is confusing. It is not explained in detail in Section 3.2.1: the sentence “We assume that the forward marginals are there noisy versions…” is not clear. Section A.3.2, which is where the approach should be explained, contains an obscure statement that does not clarify how the training is performed. Then, Section A.10.3 does contain the training loss for the score model, but the reason to introduce $\sigma$ is still unclear.
- In App. A.5, what does percentage mean?

The experiments presented show some improvement with respect to the baseline models, but some questions arise:
- The improvement relative to the baseline models is stronger for CIFAR-10 than it is for ImageNet64. Performance on ImageNet64 is much more indicative of performance on models that practitioners care about. In fact, Table 2 shows that the baseline OT-CFM with 16 NFE obtains FID 15.11, while OT-CFM with Score at 16 NFE (6 of those corresponding to the score model) obtains either 15.57 or 15.69 depending on the variant. Even if the score model is cheaper to evaluate, it is hard to make a case for using OT-CFM Score given that the obtained FID is higher.
- It is also surprising that the FID with 16 NFE is higher than with 12 NFE, for both score variants. This goes against the intuition developed in the paper.
Experiments on larger scale text-to-image models would help clarify the usefulness of the suggested method.

**Questions:**

- Wouldn’t it make sense to fine-tune the FM vector field such that $\frac{1}{N}v_{\theta_{FT}}(t_n,x_{t_n}) = \frac{1}{N}v_{\theta}(t_n,x_{t_n}) + \alpha_n h_{\psi}(t_n, x_{t_n})$? That way, we don’t need to evaluate $h_{\psi}$ separately from the flow matching vector field, which means that we can arguably use the correction at every single time step without incurring additional compute costs. It seems that should work well in the cases where parallel sampling works well.

---

> ### Author Response · Authors · 2024-11-20
>
> Thank you for your review and your suggestion for improving the presentation!
>
> **Theorem 3.3:**
>
> Thank you for your valuable suggestions, we changed the presentation of the Theorem in the revised version. Please refer to the general comment for additional details.
>
> **Score Model:**
>
> We are sorry for the confusion, we revised the score model section in Sections 3.2.1 and A.3.2 of the paper with further explanations about the approach including a pseudo-code, (highlighted in brown).
>
> Score models are a type of generative models that work by gradually cleaning up random noise into meaningful data through a process guided by the score function - the gradient of the log probability density. These models use varying noise levels during generation: they start with very noisy data where the score function provides stable but coarse guidance, then progressively reduce the noise while using increasingly refined score functions to sculpt the final output. Through this iterative denoising process, random noise is transformed into samples from the target distribution.
>
> By assuming the forward marginals are a noisy version of the backward marginals (with a small $\sigma$), a score model can be used to transverse between them.
>
> **Algorithm:** Score Model Training
>
> **Input:** flow model $v_\theta$, score model $h_\psi$, $\sigma$, backward marginals $\{p_{t_n}^b(x)\}_{n=0}^N$
>
> **Repeat:**
> 1. $\epsilon \sim \mathcal{N}(0,I)$
> 2. $n\sim U(\{0,1,..,N\})$
> 3. $x_{t_n} \sim p_{t_n}^b(x)$
> 4. Take gradient descent step on
> 5. $\nabla_\psi \| h_\psi(t_n, x_{t_n} + \sigma\cdot \epsilon) + \epsilon\|^2$
>
> **Until:** converged
>
> **Appendix A.5:**
>
> Figure 7 shows a histogram of the differences in VGG features. These features are computed for uncorrected samples (flow only) and corrected samples (flow+correction models) starting from identical Gaussian noises. The $L_2$ difference between the features of the uncorrected and corrected samples is then calculated. The percentage indicates the proportion of samples with the corresponding VGG features difference. The plot demonstrates that most samples' features remain very similar, with minimal changes. The correction models only significantly modify outliers. This aligns with the observation in Figure 1 (toy problem) that correction is only needed for the symmetric difference between the Gaussian noise and $p^b_0$, which corresponds to these outliers. We have incorporated this clarification into the figure caption in the revised paper (in brown).
>
> **ImageNet-64 Results:**
>
> We had an error in the NFE count, it is fixed in the revised version, we apologize for the confusion.  The RK4 scores are counted in multiples of 4. Sampling the OT-CFM flow model with 8-RK4 steps (32 NFE) gives a score of $15.11$, 7-RK4 steps (28 NFE) gives a score of $15.71$ and  24-RK4 steps (24 NFE) gives a score of $16.60$. In contrast, sampling the OT-CFM flow model with 5-RK4 steps and 6 C-NFE (for a total of 26 NFE) results in a score of $15.57$ or $15.69$ (depending whether is was approximated). In any case, our work focuses on the lower-NFE regime where it shows the most improvement, and is the more interesting challenge.
>
> Additionally, the CIFAR-10 models were easier to train as it is a smaller dataset.  We believe that with more compute the ImageNet-64 models could perform better.
>
> *It is also surprising that the FID with 16 NFE is higher than with 12 NFE* - some correction steps are more beneficial than others, as seen in Fig.5. In general, taking correction steps reduces the FID score compared to not taking them. It is interesting that omitting some correction steps yields a slightly superior result.  We assume this happens due to errors in the correction model accumulated over the trajectory. Designing a more sophisticated correction model might be able to resolve these issues.

---

> ### Author Response · Authors · 2024-11-20
>
> **Larger Scale Text-to-Image Models:**
>
> We agree that results on large scale text-to-image models are very impactful, but we simply do not have the (very extensive) compute resources to carry out such experiments. We kindly refer the reviewer to some recent studies on flow matching such as [A,B], which were all very impactful, yet also did not include large scale text-to-image experiments, and hope that the reviewer agrees that our work can be evaluated within the (not so small) compute budget that we have.
>
>
> **"Parallel" Network:**
>
> This is a great idea! It would leverage parallelization more effectively, though it requires additional flow model training. As this approach would bake in a specific correction configuration, adapting to new configurations would necessitate further fine-tuning. It would be interesting to investigate it as a future work.
>
> **References:**
>
> [A] Alexander Tong, Kilian FATRAS, Nikolay Malkin, Guillaume Huguet, Yanlei Zhang, Jarrid Rector-
> Brooks, Guy Wolf, and Yoshua Bengio. Improving and generalizing flow-based generative mod-
> els with minibatch optimal transport. Transactions on Machine Learning Research, 2024.
>
> [B] Yaron Lipman, Ricky T. Q. Chen, Heli Ben-Hamu, Maximilian Nickel, and Matthew Le. Flow
> matching for generative modeling. In The Eleventh International Conference on Learning Representations, 2023.

---

> > ### Comment · Reviewer_k9VA · 2024-11-26
> > **Reply to authors' response**
> >
> > I appreciate the effort of the authors, but my concerns about the work remain. I keep my score.

---

> > > ### Author Response · Authors · 2024-11-27
> > >
> > > Thank you for your response, we would appreciate if you could specify your remaining concerns.

---

### Official Review · Reviewer_ekWc · 2024-11-03

**Soundness:** 4
**Presentation:** 1
**Contribution:** 2
**Rating:** 5
**Confidence:** 3

**Summary:**

The paper introduces a method to correct the path trajectories of continuous normalizing flow models trained with flow-matching. Instead of focusing on improving ODE solvers for potentially poorly fitted models, as much of the existing literature does, this work aims to train networks that directly guide the sample trajectory by correcting the model-fit failures at each step with a (residual) network $h_\theta$.

As the authors note, although the general idea has been previously explored for continuous normalizing flows [A], this paper differentiates itself by focusing on frameworks that train normalizing flows using conditional flow matching.

I have two main concerns:

1. **Theoretical Presentation**: Some theoretical sections are presented poorly and the notation could be improved. Portions of the theory seem inconsequential and do not tie back to the story (Separation of A and B in Theorem 3.3), and could be safely removed, moved to the appendix, or generally improved for easier readability. Above that, I do have some concerns about the assumptions for one of the theorems.

2. **Lack of Evidence in Large NFE Settings**: The paper initially distinguishes between improving flow matching models by $(i)$ improving model fit and $(ii)$ refining the ODE solver. However, the authors do not provide evidence that their method is beneficial in settings with a large NFE. Specifically, if the generative performance is poor due to model-fit issues, merely increasing NFEs should not significantly help; however, the results show that increasing NFE without any correction is still getting the same performance as doing correction but with lower NFE. Therefore, to convincingly advocate for their method, experiments in large NFE settings are essential.

The second concern is crucial for connecting the results back to the paper's motivation and introduction. Indeed, the paper even has a line (L142) mentioning that "while a larger $N$ reduces discretization error, it also leads to a greater accumulation of prediction error", so why do we not have comparisons in precisely these large NFE settings?

My interpretation based on the experimental results is that adding the guidance $h$ seems to be more similar to distillation methods that allow the model to take larger steps, as opposed to inherently fixing model-fit errors. I also have some more detailed points and a suggested ablation which I will present in the "questions" section.

All in all, the paper does seem to introduce a nice method at its core, but without sufficient experimental evidence and improvements in writing, it is not ready for acceptance!

**Strengths:**

1. The paper aims to address the problems associated with image generation in possibly the most performant variant of generative models. Any improvements to flow matching approaches are therefore highly useful!

2. The background, introduction, and discussion of related work are thorough and clear.

3. The main idea is clearly presented with a nice figure that helps understanding.

4. The theoretical contribution appears sound, though not all of them directly factor into the methodology.

5. I appreciate the amount of detail and extra context that is included in the Appendix for the ablations and comparisons.

**Weaknesses:**

1. I find that separating Theorem 3.3 into parts A and B is tangential to the story and overly confusing. In reality, we do not have full control over our correction network and its Lipschitz constant. Therefore, we can never determine the best scheduling. This section seems like its being theoretical for its own sake! It might be clearer to simply present Lemma A.2 of the appendix in its most general form:
$$W_2(p^b_t, p^f_t) \le W_2(p^b_{t_0}, p^f_{t_0}) \cdot e^{L(t-t_0)}$$
and say that improving the Wasserstein distance on the RHS for $t_0$ can effectively bound the Wasserstein distance on the LHS, especially for $t$ that is sufficiently close to $t_0$. I don't think A, B, and examples 3.4 and 3.5 are particularly insightful when it is not directly factored into the decisions made in the experiments. The results in A and B can still be included, but in the appendix.

2. The parallel sampling section seems slightly oversold! To my understanding, while both forward passes can be done in parallel, it cannot be done in one batch because the forward call is on different methods. Could you please provide a time comparison between parallel and serial sampling on one experiment with the hardware that you have?

3. The statement of Lemma 3.6 seems to spill over to the rest of the main text and I generally do not agree with the base assumption that $p_t^f = p^b_{t, \sigma}$ which is the main driver for Lemma 3.6. Please let me know if I am misunderstanding this!

4. I don't find the comparison between this method and Dai and Wipf [B] appropriate! [B] trains a VAE on VAE to fix problems associated with the dimensionality mismatch between the data manifold and the manifold induced by the (first) VAE. That is not a concern in flow-matching and diffusion models as these models are known not to suffer from the manifold mismatch difficulties as much.

5. Although FIDs are still being widely used for evaluation, there have been clear flaws associated with them and the simplistic Inception network [C]. Please use DinoV2 Frechet Distances for the comparisons from [C], in addition to the widely used FID metric.

6. Please also provide evaluations "matching" the same NFEs in the corresponding non-corrected models.

### Minor points

1. I personally do not agree with the notation abuse of rewriting the conditional probability flow $p_t(x | z)$ as the marginal probability flow $p_t(x)$; it is highly confusing in my opinion.

2. Rather than introducing the new probability flows $\nu_t$ and $\mu_t$, in theorem 3.3, please consider using the same $p^b_t$ and $p^f_t$ for reduced notation overhead, and then restate the theorem in full formality for the appendix.

3. (nitpick) In Eq. (8), $t$ should be a sub-index of $u$.

**Questions:**

1. Instead of correcting the marginals, what happens if you simply define a new network $\nu_\theta(t, x) := h_\theta(t, x) + v(t, x)$ and fine tune using the CFM loss? That seems like a more direct way to fix prediction error, so I'm curious if there is any gain from using the relatively more complicated algorithm you have suggested.


### References

[A] Chen Xu, Xiuyuan Cheng, and Yao Xie. Normalizing flow neural networks by jko scheme. Advances
in Neural Information Processing Systems, 36, 2024.

[B] Dai, Bin, and David Wipf. "Diagnosing and enhancing VAE models." arXiv preprint arXiv:1903.05789 (2019).

[C] Stein, George, et al. "Exposing flaws of generative model evaluation metrics and their unfair treatment of diffusion models." Advances in Neural Information Processing Systems 36 (2024).

---

> ### Author Response · Authors · 2024-11-20
>
> We would like to thank the reviewers for their detailed and constructive feedback.
>
> **Presentation:**
>
> Thank you for your presentation suggestions, we incorporated them in the revised paper. Please refer to the general comment for additional details.
>
> **Theorem 3.3:**
>
> Thank you for this point. The relationship between the theory and algorithm is explained in detail in the general comment.
>
> **Larger NFEs:**
>
> Thank you for raising this point, we apologize for the confusion. In the sentence ”while a larger reduces discretization error, it also leads to a greater accumulation of prediction error” - by greater accumulation we meant that the sum of the prediction error is over more steps, and not necessarily that the prediction error is greater as it normalized by the number of steps, but to the best of our knowledge there is no way of measuring it.  We agree that this is not phrased well and confusing and we removed it from the revised paper.
>
> Generally, achieving low FID with a low number of NFEs is always preferred over a high number of NFEs.  In our case,  sampling with a large number of NFEs will require training and evaluation over more marginals up to approximations - marginals interpolation or using only a subset of the marginals.  This makes the training and evaluation more difficult and therefore we did not add it to the paper. While this is worthy of future work, it is beyond the scope of this paper
>
> However, we understand that it is one of the reviewer's main concerns and in order to show evidence in the large NFEs settings within the limited discussion period time we made an approximated evaluation. With the trained correction models (score and classifier) over 10-RK4 steps (40 NFEs) employed in the paper, we evaluated a setting where the base flow model (OT-CFM [A]) is sampled with 50-RK4 steps (200 NFEs), which is a substantially larger number of NFEs. This measures the performance in large NFE setting, but also generalization of the correction models, as they were not trained over these marginals which are more refined, as the ODE-solver is integrated over a larger number of steps. The FID score of 50-RK4 steps of OT-CFM is 3.67.  The evaluation of correction step-sizes of the classifier and score correction models are a normalized version (scaled down) of the ones used in the paper, as the correction is more subtle. In the table below are the results.
>
> | *Correction Model* | NFE ↓ | C-NFE ↓ | OT-CFM FID ↓ |
> |-------------------|-------|----------|--------------|
> | No Correction | 200 | 0 | 3.67 |
> | Classifier | 202 | 1 *($n=8$)* | 3.44 |
> | | 206 | 3 *($n=8,9,10$)* | 3.46 |
> | | 210 | 5 *($n=6,7,8,9,10$)* | 3.52 |
> | | 222 | 11 *($\forall n$)* | 3.54 |
> | Score | 201 | 1 *($n=10$)* | 3.64 |
> | | 203 | 3 *($n=0,5,10$)* | 3.54 |
> | | 211 | 11 *($\forall n$)* | 3.51 |
>
> The results show that it always beneficial to use correction.  However, adding more correction steps for the classifier model was not beneficial, we believe this is due to the generalization nature of this experiment. It would be interesting to explore the generalization capacities of the correction models in future work.
>
> **Parallel Sampling:**
>
> The reviewer raises a fair point. You are correct that the forward passes for the two models cannot be executed in a single batch, since they involve different method calls. The parallel sampling is parallel since no model is dependent on the output of the other model, and therefore it is possible to parallelize.
>
> Cuda has a queue, where the CPU sends tasks to be run on the GPU. The GPU  may execute tasks in parallel that are independent of one another, such as our parallel sampling. The CPU may wait when the queue is full or during synchronization events, (i.e. item(), synchronize(), etc). Given a powerful enough Cuda machine, the models could be run in parallel on different GPUs or using advanced parallelism techniques. Parallel calculation on separate devices could be useful when the correction calculation takes more time than the overhead of moving between devices, as is the case for large models.
>
> The extent of parallel processing viability will depend on the specific hardware and infrastructure available. However, on our machines, it is not possible due to the hardware limitations. Nevertheless, we want to emphasize that this approach is parallelizable in principle. We will add this explanation in the Appendix (A.3.1 Parallel Sampling) of the revised paper (in brown) and tone down the presentation.

---

> > ### Author Response · Authors · 2024-11-20
> >
> > **Assumption in Lemma 3.6:**
> >
> > The main motivation for Lemma 3.6 is to illustrate why using a single time-step Score model correction could yield a reduction in the 2-Wasserstein distance between forward and backward marginals. As we show, under some assumption, the 2-Wasserstein distance diminishes.  Therefore, taking the correction step makes sense. The "proof" of why it works is in the experiments.
> >
> > That said, we acknowledge this is an assumption, and to relax it, a classifier model can be utilized instead. This alternative approach does not require the forward and backward marginals to have the same underlying structure.
> >
> > **Comparison with  Dai and Wipf:**
> >
> > The connection is in training an additional model for the correction of the initial distribution of the original model, which is similar in high level. We changed the explanation for it in the revised paper (in brown) to make it more precise.
> >
> > **DinoV2 FD:**
> >
> > Thank you for bringing this metric to our attention. We are happy to add such an evaluation to the final version of the paper.
> > We are in the progress of evaluating with this metric, however we might not be able to complete the evaluation during the discussion period due to limited resources.
> > Even though there are better metrics than FID, and we agree that it's important to add this comparison, we posit that our results using FID are still relevant. Most of the research in diffusion models over the last 5 years has been based on FID, and clearly there has been steady progress. So while our evaluation metric can be improved, we believe our contribution can be fairly evaluated for the conference even with our current FID scores.
> >
> > **Evaluations on Flow Models:**
> >
> > Thank for this suggestion. As 1 RK4 step requires 4 NFEs, we can only "match" the results up to 4 steps with the base flow models. We extended the results in Appendix A.6.6 (Flow Models RK4 Results) in the revised paper (in brown), the caption of Table 1 references it. For additional details please refer to the general comment.
> >
> > **Vector Field Correction Network:**
> >
> > This is a good question. The network the reviewer suggested will adjust the vector field itself rather than the sampled trajectory within the same vector field. We are in the progress of testing your suggestion and will attach the results soon.
> >
> > **Minor Points:** Thank you for your attention to detail- we changed them in the revised version (in brown).
> >
> > **References:**
> >
> > [A] Alexander Tong, Kilian FATRAS, Nikolay Malkin, Guillaume Huguet, Yanlei Zhang, Jarrid Rector-
> > Brooks, Guy Wolf, and Yoshua Bengio. Improving and generalizing flow-based generative mod-
> > els with minibatch optimal transport. Transactions on Machine Learning Research, 2024.

---

> > > ### Author Response · Authors · 2024-11-23
> > > **Vector Field Correction Network Results**
> > >
> > > **Vector Field Correction Network:**
> > >
> > > We implemented the suggested experiment with both flow models as OT-CFM after experimenting with different hyper-parameters: batch size (64/128), lr (2e-4,1e-4,2e-5), lr-scheduler (none or LambdaLR), the number of warmup steps for the lr-scheduler (none or 5000), and CFM or OT-CFM for the correction model. The model with the best FID score had the same hyper-parameters as the original OT-CFM model. The following are the results obtained by sampling the models using the same number of steps:
> > >
> > > | RK4-steps | NFE | FID ↓ |
> > > |-----------|-----|-------|
> > > | 10        | 40  | 4.44  |
> > > | 11        | 44  | 4.09  |
> > > | 12        | 48  | 3.79  |
> > > | 13        | 52  | 3.56  |
> > >
> > > Our score model outperforms these results, we achieve a lower FID score with less NFEs:
> > >
> > > | RK4-steps | C-NFE | NFE | FID ↓ |
> > > |-----------|--------|-----|-------|
> > > | 10        | 1 *(n=10)* | 41  | 3.45  |
> > > | 10        | 3 *(n=0,5,10)* | 43  | 3.38  |
> > > | 10        | 11 *(n=0...10)* | 51  | 3.37  |
> > >
> > > This correction fits a residual model in order to obtain a "corrected" vector field. The residual model's objective focuses solely on compensating for the existing model's limitations without introducing additional information. In contrast, our correction model is trained on the backward marginals which are not available through the original loss objective. In general, Marginal Matching can be implemented on top of this corrected vector field as the idea is orthogonal to ours and fundamentally distinct.

---

> ### Comment · Area_Chair_KgqQ · 2024-11-26
> **reply author response**
>
> As the rebuttal deadline is approaching, please acknowledge the response from the authors and reply if needed.

---

> ### Comment · Reviewer_ekWc · 2024-11-27
> **Thank you!**
>
> Thank you very much for the response!
>
> 1. I do like the theoretical section much more now. Thank you very much for the edits!
> 2. I especially appreciate you implementing the vector field correction approach. Please add that to the final version of the paper; I think it is a very relevant ablation. I would even recommend tuning that approach more.
> 3. I look forward to the DinoV2 results in the final iteration of the paper. The paper that originally proposed this showed that diffusion models don't get good FID scores sometimes compared to GANs, even though they obviously outperform them from a qualitative perspective. One thing I noticed from all of your results is that your improvements are really marginal, and as a practitioner, it might not be worth all the additional effort to fine-tune or train my model using your approach. The DinoV2 results might even help your method to stand out more if it is truly improving generation quality.
> 4. Finally, and this is perhaps a more philosophical concern, but the line between ODE discretization error and model-fit error is much blurrier than you explain in the paper. You can, in principle, many times sidestep the model-fit error if you increase the resolution of your ODE solver. Let me give you a hypothetical example: let's say your model learns the true vector field with a an error that comes from a zero-center distribution. Now, if you increase the discretization steps, the total error rate converges to 0, just as a result of the law of large numbers. I'm obviously not suggesting this case happens in practice, but I think the arguments made between the "model-fit" vs. "discretization" error should be presented in a more convincing manner.
>
> Overall, I would like to increase my score for all the effort the authors put into the rebuttal! But I still have some reservations.
>
> In any case, looking forward to seeing the final version of your paper :)

---

> > ### Author Response · Authors · 2024-11-29
> >
> > Thank you for your time and your feedback. We would appreciate if you could be more specific regarding your concerns from raising the score.

---

### Official Review · Reviewer_iSNQ · 2024-11-04

**Soundness:** 3
**Presentation:** 4
**Contribution:** 2
**Rating:** 6
**Confidence:** 4

**Summary:**

This paper proposes the refinement of the flow-based/diffusion models with marginal matching, which delves into the gap between the forward and reverse process to mitigate the *prediction error* of the model, which is the gap between the optimal flow $u*$ and the estimated $v_\theta$. The principle behind the marginal matching is that since the model cannot be learned perfectly and the forward and backward marginal (path) must have some gap, there should be some corrector to move from the forward to the (correponding) backward marginal.
To implement the marginal matching, the paper piloted two models: score-based and robust classifier. Score-based marginal matching assumed the forward marginals to be the noisy version of the backward marginal, and the robust classifier first constructs the classifier discriminates the forward-marginal particle to the backward-marginal particle (using both data) and get the gradient to move from the forward to the backward marginal trajectory.
With this corrector model with marginal matching, the generation quality from the image datasets are improved with a few steps.

**Strengths:**

* The idea of marginal matching correction is sound, and can be ubiquitously applied to other generative model approaches.
* The convergence property from the marginal matching is concretely headlined with Wasserstein distance bound.
* Moreover, the interpolation part which says that using the data and approximate source rather than the full trajectory for training resolves some of my interests, which is the multi-marginal matching from the partial trajectory rather than the full trajectory.

**Weaknesses:**

* My main concern is that the idea of the marginal matching correction is not directly used in the score model, as the forward marginal is not directly connected to the backward marginal but is considered as the noisy version of the backward marginal. If so, (as I mentioned in the question) the corrector part is the external-model version of the existing work. As this paper implies that the score version of the corrector works better than the robust classifier version, it would be better if this issue is resolved.

**Questions:**

* For my understanding, the corrector model $h_\psi$ constructed by the score model is the same or similar to the predictor-corrector proposed in [Song et al, 2021]? If it is so, the score model part of the marginal matching is the flow-based equivalent of the existing work with more theoretical background with Wasserstein distance bound and external corrector model.
* The result in Table 1 will be more concrete if the RK4 solver includes NFE=24, 28, and 32 (as a single step uses 4 NFE). I'm curious that the paper mentioned that the NFE of a single RK4 step is 2
* How is the corrector step $\alpha$ chosen? This seems that the choice of $\alpha$ heavily affects the performance of the corrector. Are the hyperparameters in Table 10-12 hand-crafted?

```
[Song et al., 2021] Score-Based Generative Modeling through Stochastic Differential Equations, ICLR 2021
```

---

I recommend borderline acceptance in the current stage of review: this can be varied in the discussion period, and additional question can be added.

---

> ### Author Response · Authors · 2024-11-20
>
> We thank the reviewer for the valuable input. We address each point separately:
>
> **Score Model:**
>
> The corrected inference algorithm is a predictor-corrector algorithm, however the score model and the sampling is different from the one used in [Song et al, 2021].  Below are the main differences:
>
>
> * Model Training: The score model used in [Song et al, 2021] is trained to predict the score function of the data distribution with varying degrees of Gaussian noise (added by the SDE/ODE). In contrast, our score model learns to fit the backward marginals distributions trajectory obtained by integrating over the flow model in reverse. We added a pseudo-code for training the correction score model in the revised paper under the Score Model section (3.2.1).
>
> * Time-Dependency: In [Song et al, 2021] the time $t_n$ refers to different noise levels, where in our model the noise level is constant and it refers to the time of the $n^{th}$ step of the flow model's ODE-solver where the *single* correction step is applied.
>
> * Predictor-Corrector Distribution: In [Song et al, 2021] the same model is used for the predictor and the corrector sampling in order to reduce the error in the approximation of the distribution this model represents. In our work the predictor is a flow model and the corrector can be a score model.  The correction and the flow model approximate different distributions.  The correction model is trained after the flow model with the aim of covering its prediction error to gain superior performance (see Appendix A.10.3).
>
> * Corrector Sampling: In [Song et al, 2021] the corrector uses Langevin dynamics with update step: $x_{t_n}+\alpha_n h_\psi(t_n,x_{t_n})+ \sqrt{2\alpha_n} \epsilon$. Conversely, in our algorithm the correction step is: $x_{t_n}+\alpha_n h_\psi(t_n,x_{t_n}+\beta_n \epsilon)$. Sampling flow models is done without intermediate noise, adding noise to this process is not advisable. In practice, we experimented with Langevin dynamics and noticed that it yields inferior results. Additionally, only one correction step is taken at each time-step, which is very different from how Langevin Dynamics is used.
>
>
> The marginal matching is always used as the score model is trained to fit the backward marginals distributions. The score model directs the samples to the matching time backward marginals, thereby applying it results in a better alignment. This is the reason we expect the score model to improve the performance.
>
>
> **Table 1:**
>
> You are correct, thank you for pointing it out and we apologize for the confusion, it is fixed in the revised version. The additional results are added to an extended table in the appendix due to lack of space, a reference to this table is added in the caption of Table 1 in the revised version. For additional information please refer to the general comment.
>
> **Corrector-Step**:
>
> Thank you for raising this point. The hyper-parameter $\alpha$ was explored through a grid search.  Please refer to the general comment for additional details.

---

> > ### Comment · Reviewer_iSNQ · 2024-11-26
> >
> > Thank you for the response. Based on this rebuttal and the rebuttals of other reviewers, I maintain the rating in the current stage of review.

---

### Official Review · Reviewer_ubfw · 2024-11-04

**Soundness:** 2
**Presentation:** 2
**Contribution:** 3
**Rating:** 5
**Confidence:** 3

**Summary:**

This paper proposes using corrector steps during the sampling of flow models. Two correction models are proposed: a score model, and a classifier model. If I understand correctly, in practice, the score model is trained with a constant noise instead of being time-dependent. The classifier model should act similarly to a discriminator, but empirical performances seem to favor the score model.

**Strengths:**

- Discusses using additional inference-time compute which could be interesting.

**Weaknesses:**

- The concrete connection between theoretical derivation and proposed algorithm is unclear to me.
- There is also missing information regarding the method.

**Questions:**

- How does Algorithm 1 relate to the idea of the backward marginal?

- Why does Algorithm 1 add noise in line 4? Doesn't this add bias to the sampling when no correction is used?

- How do you choose beta and alpha?

- The proposal for the score model is to a time-dependent score. But in practice, only a constant noise is used. Why is that?

- If you used a time-dependent score and set beta = sqrt(2 * alpha), does your method recover Langevin dynamics as the corrector step?

- What is "Backward Marginals Interpolation"? How is this different from the standard "Score" correction model?

- Why can't this be used with diffusion model sampling (by just replacing line 6)? Intuitively, the choice of corrector steps should be independent to the predictor (ODE/SDE) steps.

---

> ### Author Response · Authors · 2024-11-20
>
> Thank you for your feedback and time.
> Here are our responses:
>
> **Connection between Theory and Algorithm:**
>
> Thank you for raising this point, for an elaboration on the connection between the theory and the proposed algorithm, please refer to the general comment.
>
> **Algorithm 1:**
>
> *How does Algorithm 1 relate to the idea of the backward marginal?*
>
> The backward marginals are manifested in the training process of $h_\psi$, the correction model, which learns to align the forward with the backward marginals.  We added a pseudo-code for training the correction score model in the revised paper under the Score Model section (3.2.1), for more details about training the correction models see A.3.2 and A.3.3.
> Algorithm 1 is the corrected inference time algorithm and assumes it has access to $h_\psi$. On lines $5$ and $8$ in the algorithm correction steps are taken.
>
>
> *Why does Algorithm 1 add noise in line 4? Doesn’t this add bias to the sampling when no correction is used?*
>
> Thank you for this question! You are correct and we did not add any noise to the sample when no correction steps were taken. We fixed the pseudo-code in the revised version, the changes are highlighted in brown, we apologize for the confusion.
>
> *How do you choose beta and alpha?*
>
> Thank you for raising this question. Alpha and beta are hyper-parameters that were explored with hyper-parameter tuning - using a grid search of the values. Please refer to the general comment for additional details.
>
>
> **Score Model:**
>
> *The proposal for the score model is to a time-dependent score. But in practice, only a constant noise is used. Why is that?*
>
> This is a great question. The time-dependency $t_n$ of the score model $s_\psi(t_n, x_{t_n})$ refers to the $n^{th}$ step of the flow model's ODE-solver where the correction $s_\psi$ is applied. This is in contrast to the time-dependency in the original score model which refers to the noise level. The score model is trained with a constant value of noise with time dependency over the ODE-solver time-steps.
>
> In Appendix A.3.2 (line 1126) we elaborate on the choice of a score model with a constant noise level.  We emphasize that while more sophisticated approaches like annealing score models with time-varying $\sigma$ exist, they were unsuitable for our needs.  This is because our inference process that starts at a forward marginal or an intermediate point where we lack information about the current and next noise levels.  Therefore, we cannot use these models effectively, as they assume knowledge of the noise levels.
>
> *If you used a time-dependent score and set beta = sqrt(2 * alpha), does your method recover Langevin dynamics as the corrector step?*
>
> It would not be the same, assuming $\beta_n=\sqrt{2\alpha_n}$.  In Langevin Dynamics the update step is: $x_{t_n}+\alpha_n h_\psi(t_n,x_{t_n})+ \sqrt{2\alpha_n} \epsilon$, where in our algorithm the update step is $x_{t_n}+\alpha_n h_\psi(t_n,x_{t_n}+\sqrt{2\alpha_n} \epsilon)$.  As you mentioned before, adding noise to the sampling of the vector field adds bias as sampling flow models is done without intermediate noise. In practice, we experimented with Langevin dynamics and noticed that it yields inferior results. Additionally, only one correction step is taken at each time-step, which is very different from how Langevin Dynamics is used.
>
> **Backward Marginals Interpolation:**
>
> *What is ”Backward Marginals Interpolation”? How is this different from the standard ”Score” correction model?*
>
> In order to keep the training simulation-free all the marginals are stored ahead of time. The ”Backward Marginals Interpolation” is an approximation method for the backward trajectory that saves storage. In the standard training of score correction model, it trains over the exact backward marginals computed with the ODE-solver. In contrast, the ”Backward Marginals Interpolation” version trains over an approximation of that trajectory.  This approximation is obtained by interpolating the data $p^b_1$ and $p^b_0$ (approximate source distribution).  In the standard case $p^b_t$ is calculated with an ODE-solver, however in the ”Backward Marginals Interpolation” case $p^b_t$ is an interpolation with coefficient $t$ (line 420 in the paper).
>
>
> **Diffusion Model Sampling**
>
> *Why can’t this be used with diffusion model sampling (by just replacing line 6)? Intuitively, the choice of corrector steps should be independent to the predictor (ODE/SDE) steps.*
>
> This is an interesting question, and we briefly mentioned it in the Conclusions and Limitations section.  In general, diffusion models are "approximately invertible" where flow models can be assumed to represent an invertible vector field as they are generally straighter, and our method depends on it. It would be an interesting direction to explore the extension of our method to these models.

---

### Author Response · Authors · 2024-11-20

We thank the reviewers for their valuable insights and suggestions. We have responded to each reviewer individually. In addition, we provide this general response to address points that were brought up by multiple reviewers. We incorporated all the reviewers suggestions in the revised version of the paper, the  changes and additions are highlighted in brown.

**Presentation:**

We modified the theoretical presentation in the revised paper of both Theorem 3.3 and Lemma 3.6, as suggested by reviewers ekWc and k9VA.
We added the lemma from the Appendix to the main paper and changed Theorem 3.3 and Lemma 3.6 to be informal in the main paper with the formal versions in the appendix. The informal version Theorem 3.3 does not contain parts A and B.

**Hyper-parameters:**

Alpha and beta are hyper-parameters of the algorithm, tuned through grid search - full details of this search are provided in Appendix A.10.4 of the revised paper, as requested by reviewers ubfw and iSNQ.

*Noise $\beta$:* The configuration with the best FID was selected from a grid search over the interval $[0,0.1]$ with a step size of 0.01.

*Step size $\alpha$:* The configuration with the best FID was selected from a grid search over the interval $[0,2]$ with a step size of 0.05. For the classifier's final step, the grid search was conducted over the interval $[0,0.1]$ with a step size of 0.01.

The step-size heavily affects the performance but it is generally not too sensitive to it.  We tested the sensitivity of our correction models by adjusting their step sizes up and down by $10$%. For instance, where the original step size was 1.0, we tested with both 0.9 ($10$% decrease) and 1.1 ($10$% increase). Here are the results we observed:

| *Correction Model* | C-NFE ↓ | mean FID | std FID |
|-------------------|----------|-----------|----------|
| Classifier | 1 *($n=8$)* | 3.58 | 0.015 |
| | 5 *($n=6,7,8,9,10$)* | 3.49 | 0.011 |
| Score | 1 *($n=10$)* | 3.46 | 0.01 |
| | 3 *($n=0,5,10$)* | 3.40 | 0.017 |

The mean and std is computed over all the FID results from the different configurations obtained by the step-size adjustment described above. As shown in the table, the mean closely matches the result reported in the paper and exhibits minimal variance. The performance remained largely consistent across different step sizes.

**Connection between Theory and Practice:**

Following the questions of reviewers ubfw and ekWc we emphasize that the Theorem 3.3 as well as the examples (linear and multiplicative reductions) fit the idea of the paper and are important to our story as they motivate the use of different correction steps as illustrated in Figure 5. Figure 5 demonstrates that some correction steps exert more influence on the performance than others. This supports the findings of the theorem as it shows that depending on the error reduction, different correction steps influence the reduction to the final bound differently. While the exact error reduction is unknown in practice, this theorem explains why different time-step correction behave differently and motivates us to seek the influence of different time-step corrections, which can be determined empirically. Additionally, this theorem illustrates why the corrected inference algorithm is sound in principle.

---

> ### Author Response · Authors · 2024-11-20
>
> **Extended Table 1:**
>
> We added additional results to the "Flow models RK4 Results" table in Appendix A.6.6 and reference it from the caption of Table 1 in the main paper, as requested by reviewers iSNQ and ekWc.  These are the results:
>
> | Model       | RK4-steps | NFE ↓   | FID ↓   |
> |-------------|-----------|---------|---------|
> | OT-CFM [A]  | 10        | 40      | 4.34    |
> |             | 11        | 44      | 3.96    |
> |             | 12        | 48      | 3.73    |
> |             | 13        | 52      | 3.59    |
> |             | 14        | 56      | 3.52    |
> |             | 15        | **60**  | **3.47**|
> |             | 20        | 80      | 3.48    |
> |             | 40        | 160     | 3.65    |
> |             | 50        | 200     | 3.67    |
> |             | 100       | 400     | 3.69    |
> | I-CFM [A]   | 10        | 40      | 4.29    |
> |             | 11        | 44      | 3.96    |
> |             | 12        | 48      | 3.74    |
> |             | 13        | 52      | 3.60    |
> |             | 14        | 56      | 3.52    |
> |             | 15        | **60**  | **3.47**|
> |             | 20        | 80      | 3.47    |
> |             | 40        | 160     | 3.63    |
> |             | 50        | 200     | 3.64    |
> |             | 100       | 400     | 3.66    |
>
> ---
>
> | Ours        | NFE ↓ | C-NFE ↓                   | OT-CFM FID ↓ | I-CFM FID ↓ |
> |-------------|-------|---------------------------|--------------|-------------|
> | Score       | 41    | 1 *($n=10$)*              | 3.45         | 3.47        |
> |             | 43    | 3 *($n=0,5,10$)*          | 3.38         | 3.39        |
> |             | **51**| 11 *($n=0\dots10$)*       | **3.37**     | **3.38**    |
> | Classifier  | 42    | 1 *($n=8$)*               | 3.57         | 3.77        |
> |             | 46    | 3 *($n=8,9,10$)*          | 3.48         | 3.67        |
> |             | **50**| 5 *($n=6,7,8,9,10$)*      | **3.47**     | **3.62**    |
> |             | 62    | 11 *($n=0\dots10$)*       | 3.48         | 3.63        |
>
>
> Correction steps improve performance with minimal additional NFEs, we outperform the baselines in the small NFE regime. With 43 (score)/46 (classifier) NFEs, our correction models achieve the same or better FID scores than flow models using 60 NFEs, representing a 23-28\% reduction. Even a single correction step significantly reduces the FID.
>
> **References:**
>
> [A] Alexander Tong, Kilian FATRAS, Nikolay Malkin, Guillaume Huguet, Yanlei Zhang, Jarrid Rector-Brooks, Guy Wolf, and Yoshua Bengio. Improving and generalizing flow-based generative models with minibatch optimal transport. Transactions on Machine Learning Research, 2024.

---

### Meta-Review · Area_Chair_KgqQ · 2024-12-16

**Metareview:**

This paper presents a method to reduce the prediction error in flow matching. The basic idea is to learn a corrector to reduce the gap between the marginal distributions associated with the forward and backward trajectories. Theoretical analysis is provided for the overall framework and two practical algorithms are presented. Several reviewers raise questions on the theoretical result and its connections to the algorithms. They also provide suggestions to improve the presentation. Some reviewers think the empirical results are not sufficient to demonstrate the advantages of the proposed method. There exist gaps between theory and algorithms. Some assumptions appear to be strong. For instance, the assumption in Section 3.2.1. The authors are encouraged to improve their results and presentation following the reviewers’s comments.

**Additional Comments On Reviewer Discussion:**

The reviewers raise some questions on the results and presentations. The authors reply by modifying the paper and adding clarifications in the response. Most reviewers are not convinced and keep their original evaluation of this work.

---

### Decision · Program_Chairs · 2025-01-22

Reject